# Discovery and characterization of submarine groundwater discharge in the Siberian Arctic seas: A case study in the Buor-Khaya Gulf, Laptev Sea

Alexander N. Charkin[1,2], Michiel Rutgers van der Loeff[3], Natalia E. Shakhova[2,4], Örjan Gustafsson[5], Oleg V. Dudarev[1,2], Maxim S. Cherepnev[2], Anatoly N. Salyuk[1], Andrey V. Koshurnikov[6], Eduard A. Spivak[1], Alexey Y. Gunar[6], Alexey S. Ruban[2], Igor P. Semiletov[1,2,4]

[1]Pacific Oceanological Institute (POI), Far Eastern Branch of Russian Academy of Sciences Russian Academy of Sciences (FEBRAS) , Vladivostok, Russia
[2]National Research Tomsk Polytechnic University, Russia
[3]Alfred-Wegener Institute, Helmholtz Center for Polar and Marine Research, Bremerhaven, Germany
[4]International Arctic Research Center (IARC), University of Alaska, Fairbanks, USA
[5]Dept. of Environmental Science and Analytical Chemistry, and the Bolin Centre for Climate Research, Stockholm University, Stockholm, Sweden
[6]Moscow State University, Russia

*Correspondence to:* Alexander N. Charkin (charkin@poi.dvo.ru)

**Abstract.** It has been suggested that increasing terrestrial water discharge to the Arctic Ocean may partly occur as submarine groundwater discharge (SGD), yet there are no direct observations of this phenomenon in the Arctic shelf seas. This study tests the hypothesis that SGD does exist in the Siberian-Arctic shelf seas, but its dynamics may be largely controlled by complicated geocryological conditions such as permafrost. The field-observational approach in the southeast Laptev Sea used a combination of hydrological (temperature, salinity), geological (bottom sediment drilling, geoelectric surveys), and geochemical ($^{224}$Ra, $^{223}$Ra, $^{228}$Ra and $^{226}$Ra) techniques. Active SGD was documented in the vicinity of the Lena River delta with two different operational modes. In the first system, groundwater discharges through tectonogenic permafrost talik zones was registered in both winter and summer. The second SGD mechanism was cryogenic squeezing out of brine and water-soluble salts detected on the periphery of ice hummocks in the winter. The proposed mechanisms of groundwater transport and discharge in the arctic land-shelf system is elaborated. Through salinity versus $^{224}$Ra and $^{224}$Ra/$^{223}$Ra diagrams, the three main SGD-influenced water masses were identified and their end-member composition was constrained. Based on simple mass balance box models, discharge rates at site in the submarine permafrost talik zone were $1.7 \times 10^6$m$^3$d$^{-1}$ or 19.9m$^3$s$^{-1}$, which is much higher than the April discharge of the Yana River. Further studies should apply these techniques on a broader scale with the objective of elucidating the relative importance of the SGD transport vector relative to surface freshwater discharge for both water balance and aquatic components such as dissolved organic carbon, carbon dioxide, methane, and nutrients.

## 1 Introduction

The Arctic system constitutes a unique and important environment with a central role in the dynamics and evolution of the earth system. Global warming has regional effects on the Arctic, including on all cryospheric features. Energy and water fluxes shape the regional temperature regime, which is a primary factor in determining the physical state

(frozen vs. thawed), trace gas fluxes, rates of productivity, and the link to regional climate (Serreze et al., 2006; Vörösmarty et al. 2001; Shakhova et al., 2010, 2014, 2017; Semiletov et al., 2007, 2016; Vonk et al., 2012).

The Arctic is inherently a highly dynamic system. Yet there is mounting evidence that it is now experiencing an unprecedented degree of environmental change. Many of these changes are linked to the Arctic hydrologic cycle. The delivery of freshwater (FW) from the continental land mass is of special importance to the Arctic Ocean; the Arctic Ocean contains only 1% of global ocean water, yet it receives 11% of global river runoff (Shiklomanov et al., 2000). The Arctic Ocean is the most river-influenced and landlocked of all oceans and is the only ocean surrounded by permafrost, with a drainage basin area greater than its surface area (Semiletov et al., 2000, 2012; Vörösmarty et al., 2001; Macdonald et al., 2008). Annual FW inflow contributes as much as 10% of the FW in the upper 100m of the entire Arctic Ocean (Barry and Serreze 2000). Approximately three-quarters of Arctic Ocean riverine FW input derives from the Eurasian portion of the Arctic Ocean watershed, and three rivers (the Yenisei, Lena, and Ob rivers) are responsible for approximately 70% of this contribution (Carmack, 2000; Gordeev et al., 1996). This water exerts a tremendous influence on the Arctic Ocean, and especially on the Eurasian shelf seas (the Barents, Kara, Laptev, and East Siberian seas). Salinity distribution and sea ice formation are affected by continental runoff. The cumulative impact of changes in FW flux to the Arctic Ocean may exert significant control over global ocean circulation by affecting the volume of North Atlantic deep water formation (IPCC, 2013).

Since the first reports of increases in winter and total discharge for several great Eurasian Arctic rivers (Savelieva et al., 2000; Semiletov et al., 2000), many studies have reported remarkable changes in water cycle components of the northern hydrological systems, such as increases of Yenisei, Lena, and Ob river runoff and changes in seasonal discharge patterns (Peterson et al., 2002; McClelland et al., 2006; Serreze et al., 2006). Arctic and subarctic watersheds are undergoing climate warming, permafrost thawing, and thermokarst formation, resulting in quantitative shifts in surface water –groundwater interaction at the basin scale. The Lena River region is a prioritized study subject because changes in the Lena River hydrology play a significant role in feeding the Transarctic Drift through changed FW export to the Laptev Sea. The Lena River is also a natural boundary between western and eastern Siberian conditions which differ in in atmospheric circulation patterns – land hydrology – sea ice condition, tectonic structure, and biological peculiarities on the Siberian shelf (Semiletov et al., 2000). Most of the Lena River basin is underlain by permafrost, about 79% with continuous permafrost, and the remainder with discontinuous permafrost (Zhang et al., 2005), while large west Siberian rivers (the Ob and Yenisei) are situated mainly in non-permafrost and/or discontinuous permafrost.

A hypothetical scheme of submarine groundwater discharge (SGD) is shown in Fig. 1. SGD is a mixture of fresh groundwater and seawater that recirculates through the subterranean estuary as result of tides and wave action, and then discharges to the ocean (Moore, 1999). It is estimated that groundwater currently comprises almost one fourth of Yukon River water discharged to the Bering Sea, which subsequently is transported into the Arctic Ocean via Bering Strait (Walvoord and Striegl, 2007). Long-term streamflow records (>30 yrs) of the Yukon River basin indicate a general upward trend in groundwater contribution to streamflow of $0.7–0.9\% \, yr^{-1}$. Changes in the base flow, such as increases in the Yenisei River flow between 1936 and 1995 (Vörösmarty et al., 2001), are also thought to reflect increased groundwater infiltration, coupled to reductions in permafrost and an increase in active layer thickness due to warmer temperatures. Recent studies using the Gravity Recovery and Climate Experiment (GRACE) have revealed a significant increase in subsurface terrestrial water storage in the Lena basin (Velicogna et al., 2012), which will have a significant impact on the terrestrial hydrology of the region, including increased base flow and alteration of seasonal runoff.

As of today, the Arctic Ocean shelf groundwater systems have been very poorly studied and the SGD details are poorly characterized. At the same time, deep-seated subsurface aquifer groundwater systems have developed under the seafloor, which are similar to those on land (Hathaway et al., 1979; Kohout et. al., 1988; Bisson, 1994; Moore, 1999). In these systems, the migration of water infiltrating from the land is possible, as is the reverse, the migration of saline water from the submarine watershed to the land. The origin and functioning of local SGD zones on the shelf is tightly connected via structural, lithological, and geomorphological features to the shelf regions. For the Arctic shelf, with boundaries of widely developed permafrost, these subsurface land-sea interactions are complicated by the geocryological conditions. The permafrost cements rocks and forms a cryogenically-confining bed (impermeable frozen rocks) (Pinneker, 1983; Romanovskii, 1983). However, springs that exist in the eastern North Slope of Alaska are clear evidence that a groundwater flow system exists in an environment mapped as continuous permafrost (Kane et al., 2013).

Until recently, knowledge about the present thermal state of subsea permafrost was mainly based on controversial modeling results. Some authors suggest that it would take ~5-7 millennia for subsea permafrost to reach the thaw point (Romanovskii et al., 2005), meaning that the coastal subsea permafrost is still stable. Others believe that in the coastal areas of the shallow Siberian Arctic seas, where permafrost was submerged most recently, *taliks* (layers or columns of thawed sediments within permafrost) might form as the result of the combined effect of geothermal flux from fault zones, the warming effect of rivers and overlying seawater, and the already present thermokarst (Shakhova and Semiletov, 2007; Shakhova et al., 2009, 2015; Nicolsky and Shakhova, 2010; Nicolsky et al., 2012). Moreover, the first results from off-shore drilling (accomplished from the fast ice in April 2011) down to 52m below the sea floor in the southeastern Laptev Sea showed that the sediment core was entirely unfrozen, and 8-12°C warmer than an on-land sediment core obtained in the nearby Chay-Tumus borehole (Shakhova et al., 2014), confirming these authors' modeling results. Besides, a major portion (>80%) of the Russian Arctic shelf is underlain by subsea permafrost; this permafrost has been degrading at increasing rates during the last 30 years, releasing organic carbon (OC) and pre-formed methane ($CH_4$) stored within and beneath subsea permafrost (Shakhova et al., 2017). Seafloor erosion could start from initial ground ice melt followed by formation of a polygonal form of seafloor landscape. It could further, or in parallel, be affected by chemical erosion caused by salt penetration into the frozen ground followed by removal of eroded material by bottom currents in the near-shore East Siberian Arctic Shelf (ESAS) zone.

According Shakhova et al. (2017), development of taliks of different origins could lead to formation of vertical erosion channels, helping gas fronts to propagate upwards towards the seafloor. These taliks could be identified by groundwater discharge that could be manifested as large, temporally and spatially variable point sources, which could have a significant impact on the hydrological and geochemical parameters of coastal waters. Another possible mechanism for preventing taliks from freezing and/or preventing talik formation could be groundwater flow through coastal sediments, especially in the areas underlain by faults. This flow could cause formation of so-called tectono-genic taliks (Romanovskii, 1983). Moreover, Frederick and Buffett's (2015) modeling suggested that SGD may play a large role in submarine permafrost evolution and gas hydrate stability; they propose that local hydrology may control the evolution of submarine permafrost as strongly as sea level variations, for example. As a result, local hydrology may explain large differences in submarine permafrost distribution between different regions with similar paleoclimatic history (Frederick and Buffett, 2015). In addition, it was also established that the existence of a topographically driven, groundwater flow system transports heat by advection from areas of high elevation to areas of low elevation (Deming et. al., 1992), which also contributes to creating conditions conducive to the the appearance of taliks. Thus, it is possible that water flowing under the permafrost can discharge through taliks in the permafrost and ultimately into the Arctic Ocean (Kane et al., 2013).

This study employs state-of-the-art measurements of the radium isotope quartette ($^{224}$Ra, half-life = 3.66 days; $^{223}$Ra, half-life = 11.4 days; $^{228}$Ra, half-life = 5.75 years; $^{226}$Ra, half-life = 1600 years). These naturally-occurring isotopes are useful environmental tracers for measuring coastal ocean mixing rates (Moore, 1996, 2000a; Charrette et al., 2003, 2007, 2013; Hancock et al., 2006), ages of river plumes (Moore and Krest, 2004; Gu et al., 2012), estuarine residence times (Moore et al., 2006; Rapaglia et al., 2010), and as indicators of SGD (Rama and Moore, 1996; Charrette et al., 2008). One must note that this research has to date primarily been conducted in the seas and estuaries of moderate, subtropical, and tropical climates, and only rarely in Arctic seas. Work in the Arctic seas has so far largely relied on the long-lived isotopes $^{228}$Ra and $^{226}$Ra (Cochran et al., 1995; Rutgers van der Loeff et al., 1995, 2003, 2012; Kadko et al., 2008, 2009; Smith et al., 2003; Trimble et al., 2004), with a very limited number of summer studies using the short-lived isotopes $^{223}$Ra, $^{224}$Ra, and $^{222}$Rn, conducted only in the seas of the American side of the Arctic (Kadko and Muench, 2005; Dimova et al., 2015; Lecher et al., 2015).

The main objective of this paper is to search for SGD in the Siberian Arctic area assumed to be occupied by continuous subsea permafrost (Romanovskii, 1983). SGD is characterized using a combination of hydrological (temperature, salinity), geological (bottom sediment drilling, geoelectric surveys), and geochemical ($^{224}$Ra, $^{223}$Ra, $^{228}$Ra and $^{226}$Ra) parameters. We focus our detailed analysis on the near-shore zone of the Laptev Sea, the northern extension of the Lena River basin, Eurasia, and a southern end of the Lena Rift (characterized by active seismo-tectonics, Imaev et. al., 2004), an area of about 2,400,000 km$^2$ in size. Finally, a mechanism to explain SGD in this Arctic land-shelf system is proposed.

## 2 Material and methods

### 2.1 Field work

In situ observations, drilling, and sampling were performed in the western Buor-Khaya Bay of the Laptev Sea within the framework of the International Siberian Shelf Studies (ISSS, Semiletov and Gustafsson, 2009) and the ISSS extension, including subsea sediment drilling annually for five years (March-April of 2011-2015) (Fig. 2). Complex exploratory surveys were performed annually in March – April 2011 – 2015 on the southeastern Laptev Sea land-fast sea ice to drill bottom sediment (Shakhova et al., 2014, 2017). The central campaign equipment included a brand new URB - 4T drilling rig, and a transport caravan consisting of two Caterpillars, a heavy all-terrain truck, and a sled train bearing two two-story mobile houses which hosted a laboratory and accommodations. Drilling was accompanied by geoelectric survey (Koshurnikov et al., 2016). Hydrological studies were also accomplished each year (2011-2015), including taking Conductivity-Temperature-Depth (CTD) measurements (SeaBird 19Plus). The most detailed sampling for radium activities was performed in March-April 2015. The choice of sampling sites was based on knowledge gained from the drilling, oceanographic, and geoelectric data obtained in the four previous years. Additional water samples were collected in September 2013 during the "Lena Delta 2013" hydrological survey expedition on board RV *Dalnie Zelentsy* (Gonçalves-Araujo et al., 2015).

### 2.2 Drilling and on-site characterization of bottom sediments

The subsea sediment boreholes were drilled from the land-fast ice in the spring using a rotary drill rig (URB-4T) and a dry drilling technique. A metal casing was drilled through the fast ice and water column and into the sediment to prevent water from entering the borehole. Full methodology details can be found in Shakhova et al. (2014). On-site

laboratory investigation was performed immediately upon sample recovery and included determining sediment temperature and lithology.

### 2.3 Geoelectric surveys

The transient electrical method (TEM), aimed at mapping the subsea permafrost table, determining the lithological structure, and searching for taliks, was performed during each field campaign. A TCIKL-7 measuring device was used for the TEM fieldwork, as described in detail in Koshurnikov et al. (2016).

The TEM results in the area highlighted several layers with various resistivities. The boundary of these layers was

10 validated and confirmed for the first time by subsea sediment drilling. The main conduction mechanism in earth materials is electrolytic, involving ion transport by dissolved salts distributed through a complex structure of interconnected pores and fractures. The resistivity of igneous and metamorphic rocks is typically high, while sedimentary rocks are usually much more conductive. Sedimentary rock resistivity is mainly controlled by the amount of water present, the salinity (free ions) of the water, and the degree of interconnections between the pores. Clay

content can also contribute by providing an additional surface conductivity mechanism (e.g., variations in grain size and pore space geometry) (Anderson et al., 1994). Resistivity of permafrost is typically high; low resistivity indicates that the layer is in the thawed state (Palacky, 1988). Further details on the employed methodology can be found elsewhere (Koshurnikov et al., 2016).

### 2.4 Hydrological surveys

The standard observations at each station started with a CTD measurement. A Seabird 19Plus CTD was used for measuring the vertical distribution of temperature and salinity. The device was used in sounding mode. During winter campaigns (2011-2015) an ice hole was made at each station using a Jiffymotor drill. The sounding equipment was

25 lowered by hand on a cable at a speed of around 1m s$^{-1}$. At the oceanographic measurement station 1507, ADCP observations over of the speed and direction of the flow took place over 8 days. For details of the summer hydrological work during Lena Delta 2013 see Gonçalves-Araujo et al., 2015.

### 2.5 Analytical methods

### 2.5.1 Sampling and measurements of $^{224}$Ra, $^{223}$Ra, $^{228}$Ra and $^{226}$Ra isotopes

In order to detect possible SGD outputs we collected water samples to measure the short-lived radium isotope activity. Water samples (20-60L) from the surface (just under the ice) and the bottom (one meter from the bottom to avoid

resuspension) were pumped through the boreholes of the fast ice using Grundfos submersible pumps. At several stations intermediate depths were also sampled. At the shallow stations (6m or less), the water samples were taken only from the middle depth. Groundwater samples (5-20L) were collected using the same submersible pumps from the drilled bottom sediment borehole. The impermeability of the metal casing prevented seawater penetration into the borehole.

In the base camp laboratory, the water was subsequently pumped at < 1Lmin$^{-1}$ through a column of manganese-coated acrylic fiber (Mn fiber) to quantitatively capture dissolved radium (Moore, 1976; Moore and Arnold, 1996). All winter

and summer samples were passed over Hytrex cartridges with 1μm nominal pore size for removing particulate thorium (Th) isotopes, before being run through the Mn fiber. After rinsing the Mn fiber samples with Ra-free deionized water to remove sea salts, samples were partially dried and placed in a closed-loop air circulation system as described in Moore and Arnold (1996). The Radium Delayed Coincidence Counter (RaDeCC) system utilizes the difference between the decay constants of the short-lived polonium (Po) daughters of [219]Rn and [220]Rn to identify alpha particles derived from [219]Rn or [220]Rn decay and hence to determine activities of [223]Ra and [224]Ra on the Mn fiber (Moore and Arnold, 1996). After the first [223]Ra and [224]Ra measurement in the base camp laboratory, Mn fibers with radium samples were aged for 2-6 weeks to allow initial excess [224]Ra to reach secular equilibrium with the [228]Th that had also absorbed on the Mn fiber. The samples were then measured again to determine [228]Th and thus to correct for supported [224]Ra (Moore and Arnold, 1996; Moore and de Oliveira, 2008; Burnett et. al., 2008). After about 100 days the samples were counted a third time to measure [227]Ac (actinium)-supported [223]Ra, but [227]Ac activity was below detection limit and, thus, the [227]Ac contribution is assumed to be negligible. The average statistical counting error was ± 8.5 % (range =± 5.6 - 39 %) for [223]Ra and ± 1.9 % (range = ± 1.4 - 19 %) for [224]Ra. We calibrated the RaDeCC systems for [224]Ra and [223]Ra measurements using [232]Th standards as described by Moore and Cai (2013).

Water samples collected during the Lena Delta 2013 expedition were immediately passed over Mn-coated acrylic fiber and analyzed on board for the short-lived radium isotopes with RaDeCC following the same methods as described above. We only report data for [224]Ra because we were not satisfied with the calibration for [223]Ra. The analysis of supported activities in these samples was performed later in the Alfred-Wegener Institute (AWI) home laboratory in Bremerhaven, Germany. Ra was leached from the fiber with hot 6n HCl, coprecipitated as $BaSO_4$, and counted with gamma spectroscopy for [226]Ra and [228]Ra (Moore, 1984).

### 2.5.2 Calculation of SGD residence time based on radium isotopes

The groundwater discharged from bottom sediments into the water column has the same [224]Ra/[223]Ra activity ratio as does pore water; this activity ratio is defined as the initial ratio ([224]Ra/[223]Ra]init), or the ratio of [224]Ra/[223]Ra defined as $t=0$ days (Hougham and Moran, 2007; Burnett et al., 2008). This initial activity ratio reflects the constant decay of parent isotopes in the bottom sediments; when the radium-enriched pore water (groundwater) enters the seawater, the activity ratio begins to change to reflect the different decay rate constants of the two isotopes. Because [224]Ra has a greater decay constant than [223]Ra ($\lambda_{224}$=0.189 day$^{-1}$ and $\lambda_{223}$=0.0606 day$^{-1}$, respectively), the activity ratio will decrease systematically with time, providing a radium-derived age (Hougham and Moran, 2007). To carry out modeling, the radium ages must take into account the possible introduction of radium from other sources, after the first enrichment has occurred (Moore and de Oliveira, 2008; Dulaiova et al., 2006).

In order to calculate the growth of the water mass "radium ages," we used Eq. (1) proposed by Moore (2000b),

$$t = \frac{\ln\left[\frac{^{224}Ra}{^{223}Ra}\right] - \ln\left[\frac{^{224}Ra}{^{223}Ra}\right]_{init}}{\lambda_{223} - \lambda_{224}} \qquad (1)$$

where $t$ represents time (days), [224]Ra/[223]Ra] is the calculated activity ratio in the water sample, [224]Ra/[223]Ra]init is the calculated activity ratio in the radium-enriched endmember, and $\lambda_{224}$ and $\lambda_{223}$ represent the respective [224]Ra and [223]Ra decay constants of 0.189 day$^{-1}$ and 0.0606 day$^{-1}$.

### 2.5.3 SGD discharge calculation

The average radium activities of water mass end – members were used for the box model mass balance mixing calculations. The SGD discharge was calculated following Eq. (2) (Moore, 1996; Krest et al., 2000; Null et al., 2012):

$$D = \frac{(A_{box} - A_{offshore}) \cdot V_{box}}{A_{GW} \cdot \tau} \qquad (2)$$

Discharge ($D$) ($m^3 d^{-1}$) is calculated from the excess activity; $A_{box}$ is the average Ra activity in the box; $A_{offshore}$ is the offshore activity in the northwestern part of Buor-Khaya Bay (Laptev Sea); $V_{box}$ is the volume of the model box; and $\tau$ is the water residence time (days).

## 3 Results and Discussion

### 3.1 Geologic, tectonic, and geoelectric settings in context of SGD characterization

The research area was located in the eastern Laptev Sea, which has been intensively studied by the author's team in all seasons since 1994 (Semiletov et al., 1996a,b; Semiletov , 1999, Semiletov et al., 2013; Charkin et al., 2011, 2015; Shakhova et al., 2010, 2014). According to the tectonic character of the region, the study area is located in the northern part of the Verkhoyansk Fold (Kharaulakh Ridge), which apparently continues offshore and forms a pre-rift basement of the Ust Lena Rift (Fig. 3). This region historical Cenozoic events are related because the North American and Eurasian plates are separated in the Eurasian Basin by the active mid-oceanic Gakkel Ridge, resulting in partial destruction of the continental crust over vast areas (Vinogradov and Drachev, 2000; Franke et al., 2001; Drachev et al., 1998, 1999; Imaev et al., 2004). As the result of stretching during the Pliocene-Quaternary times, the graben borders formed young listric faults displacing the weathering crusts of the Neogene age, which are also well-known along the Buor-Khaya Gulf coast (Figs. 3, 4a). The stretching axis was oriented towards the northeast. These listric faults are observed in coastal outcrops from the Bykovsky distributaries in the Lena River delta to the Kharaulakh depression at a distance of more than 160km (Fig. 4a) (Imaev et al., 2004).

Seismic analysis on the Laptev Sea shelf shows that one of the most intense earthquake epicenters for the Buor-Khaya Gulf is located precisely in our study area, and it coincides with the listric faults of the Ust-Lena graben borders 20km east of the Bykovsky peninsula (Figs. 2, 3, 4a, c). According to data collected from 1960-1990 (Imaev, et. al., 2004), the listric fault displacement dominated focal earthquake mechanisms (Fig. 4c).

According to our TEM data, sharp resistivity contrasts were established in the area of geophysical stations 2 and 3. The resistivity at a depth of more than 150m falls from more than 350Ω m in the west to 80Ω m in the east, locating the listric faults (Figs. 3, 4). These TEM results agree well with data provided by Imaev et al. (2004) for this region (Fig. 4). At TEM station 2, the resistivity did not exceed 9Ω m at the 162.5m surface of the high resistance layer. This indicates the existence of open tectonogenic taliks here (Pinneker, 1983; Romanovskii, 1983), which coincide with the fault (Figs. 3, 4).

At the geoelectric sections four deposition layers were distinguished, which differ by their lithology and temperature. All sections exhibited various thicknesses; low-resistance layers were discovered in each (1.7-3.4 Ω m), which testify to the presence of unfrozen rock (Koshurnikov et al., 2016). The drilling of wells 1D-15 established that this layer consists of water-bearing sands (Fig. 4). In the surface depositions at TEM station 2, we also discovered sedimentary

strata with a higher resistivity of 8.3Ω m and a thickness of 52m; these characteristics were not found in the other sections. This stratum is probably the remainder of the Quaternary Ice Complex, which consists of loess deposits. As the result of the sea transgression, this raised area was flooded, and it was possibly in part destroyed by thermal abrasion after sea transgression. As a result, due to the melting of relict ice and covering by recent sedimentary material, this area of the bottom was made level and is not clearly reflected in the recent relief (Figs. 2, 3, 4). A similar relict of the subaerial relief is Muostakh Island, which is located on the surface of the footwall block of the neighboring listric faults (which, together with the listric faults being examined, form a listric fan). This island is currently being intensively destroyed by thermal abrasion. If this assumption is true, then one can presume that the lithological composition of the shallow elements around Muostakh Island will have a composition similar to that of the sedimentary strata in TEM column 2 with a resistivity of 8.3Ω m. According to our drilling results (4D-12), the lithological composition of the sediments is mainly silty and clayey (Fig. 4).

At three TEM stations, but not at station 2, a sharp change in resistivity can be noted, from 1.7-2Ω m in water-bearing sands to 35-80Ω m, which is characteristic of permafrost (Fig. 4) (Koshurnikov et al., 2016; Palacky, 1988). Thus, it was determined by TEM that the depth of the subsea permafrost table in the study area (except for the talik site) varies from 20 to 46m. We have also confirmed the correspondence of the subsea permafrost to this range of resistivity many times during in-phase drilling and geoelectric surveys over five years (2011-2015).

## 3.2 Features of the thermohaline water structure

The 2015 field campaigns in the northwestern Buor-Khaya Gulf were conducted during the winter low-discharge season. Due to its low winter flow, the influence of the Lena River under ice conditions is very restricted (Charkin et al., 2011). Furthermore, under the ice conditions there is no winter wind mixing, which together results in moderately and highly stratified water predominating in this region in winter (Figs. 5, 6, 7). Stratification factors, shown in Fig. 5, were calculated according to Burt et al. (2014). The mixed layer depth is divided by the water column depth; a stratification factor of one indicates a fully mixed water column. The salinity right under the ice near the delta ranged from 0.04 - 3.1, while near the bottom it was 11.9-19.1. At a distance from the delta these values varied from 5.8 to 13.9 under the ice and from 17.4 to 26.1 near the bottom (Figs. 6, 7).

In the area of stations 1516-1529, a plume of water more saline than the surrounding water was discovered near the surface (Fig. 6). Another lens of more saline water which was larger in size, around 7 km in width and more than 20 km in length, was discovered at 5-6m depth (Fig. 6). Additionally, at stations 1529 and 1520, a local freshening of 0.5-1 in comparison with the neighboring stations was discovered near the bottom (Figs. 6, 7).

The temperature regime was characterized by ubiquitously negative values in the waters. Near the Delta (at stations 1502 and 1503) under the influence of river flow, the surface water temperature was near 0°C (Figs. 6, 7). As in the case with the salinity anomalies described above, low-temperature anomalies were also discovered, which were structurally consistent with the lenses also described above. In the vertical temperature profiles, these anomalies had the clear structure of a fluid with mushroom-like forms (Fig. 7). An intrusion of low-temperature waters was seen at the seafloor in the area of station 1529, shaped by isohaline = 22.5. The temperature in this cold fluid varied from -1 to -1.1°C, while at the same time, in the remaining part of the area being studied, the value varied from -0.04 to -0.97°C (Figs. 6, 7).

This low-temperature anomaly was also observed in the middle depth of the water column in under-ice conditions in previous years. Placing these stations on the tectonic map of the region, we discovered that all these anomalies coincide

with the large faults formed by the Ust-Lena graben (Fig. 3). This connection is not evident at first glance, but having taken advantage of the opportunity to measure the activity of short-lived radium isotopes in this water, we can shed light on the genesis of this anomaly.

**3.3 Radium isotopes signals in the water column**

In the winter, the radium isotope activities and the $^{224}$Ra/$^{223}$Ra activity ratio have been measured at different water column depths to determine the genesis of the thermohaline anomalies. The data are reported in Table 1. The distribution of ex$^{224}$Ra (excess $^{224}$Ra) in the water samples ranged from 1.8 to 19dpm 100L$^{-1}$, for $^{223}$Ra from 0.14 to 0.65dpm 100L$^{-1}$(Table 1, Fig. 8). The highest activities of ex$^{224}$Ra and $^{223}$Ra (19 and 0.65dpm 100L$^{-1}$, respectively) were detected at bottom station 1529, the exact same location where the thermohaline anomalies were also detected. The above-described low-temperature, salty plume at the surface and 6m depth are thus also enriched in the short-lived radium isotopes (Table 1, Figs. 8, 9). In the river plume ex$^{224}$Ra and $^{223}$Ra activities were 9 and 0.55dpm 100L$^{-1}$, respectively. The activity of the short-lived radium isotopes also increased (ex$^{224}$Ra=10.9; $^{223}$Ra=0.44) near the ice hummock (station 1501) in the Cape Muostakh shallows. An ice hummock is a smooth hill of ice that forms under extreme weather conditions during autumn ice formation.

In the "pure" groundwater sampled from the two drilling wells, the activities of short-lived radium isotopes were ex$^{224}$Ra=207, $^{223}$Ra=7.8dpm 100L$^{-1}$ for well ID15, and ex$^{224}$Ra=224, $^{223}$Ra=8.5dpm 100L$^{-1}$ for IID14. The $^{224}$Ra/$^{223}$Ra activity ratio in the groundwater was 26.5, close to the ratio found in the strongly enriched bottom horizon of stations 1529 and 1520 (Table 1). This circumstance, and the fact that the highest activities of dissolved ex$^{224}$Ra and $^{223}$Ra isotopes were observed in the bottom horizon of the stations, clearly points to submarine groundwater discharging where the thermohaline anomalies were also discovered. At first view, such an isohaline structure may indicate the existence of vortex formations on this section; however, at a very shallow depth (8m, not taking the ice into account), the leveling of bottom relief, the stretched form and greater area of the plumes in relation to the depth of the water (Fig. 5), as well as the data on temperature and short-lived radium isotopes suggest SGD as a plausible mechanism consistent with these observations.

The signals described above point to the existence of SGD near stations 1529 and 1520 during the survey in winter (March-April 2015); this hypothesis finds support in the distribution of ex$^{224}$Ra, $^{226}$Ra, and $^{228}$Ra observed in the wider Delta area in September 2013 (Table 2, Figs. 8, 9). The highest $^{224}$Ra activity found during this summer Lena Delta survey, 92.2 ± 1.9dpm 100L$^{-1}$ (Table 2, Fig. 11), was found on Sept 1 at the bottom of station 103 (Fig. 2), just 1.5 km south of station 1529. This bottom water sample at 12m depth (salinity 15.9) also showed the highest activities of $^{226}$Ra (31±1dpm 100L$^{-1}$) and $^{228}$Ra (77±4dpm 100L$^{-1}$) observed during this cruise (Table 2). Other high ex$^{224}$Ra concentrations, in excess of 25dpm 100L$^{-1}$, were observed at two more stations sampled on September 6 and 7: 69.1±1.3dpm 100L$^{-1}$ was found at a depth of 10m (bottom depth 12m, salinity 21.8) at station Y (72°N/130°E, i.e. 23km further north), and 43.4±1.2 and 36.4±1.2dpm 100L$^{-1}$ at depths of 2.5 and 5m, respectively (salinity 21.8 and 25.2, respectively), at shallow (6m water depth) station 404 (73°N/130°E, i.e. 134km further north) (Table 2). On this last day of the cruise, strong wind had caused very high turbidity at these shallow stations, and it is possible that resuspension of bottom sediments had released pore waters causing these high values. The role of SGD can be distinguished by the ratio of short- to long-lived isotopes (Rodellas et al., 2017). Surface sediments are much more important as a source of short-lived than of long-lived isotopes because ingrowth of the long-lived isotopes is much slower. During this survey $^{226}$Ra and $^{228}$Ra depended on salinity, and anomalous activities are best distinguished in a plot of their activities as function of salinity (Fig. 10). Activities at station 103 (12m) and Y (10m) are the only ones

that stand out for $^{224}$Ra, $^{226}$Ra, and $^{228}$Ra (Table 2, Fig. 10). Station 404 (5m) is enriched in $^{224}$Ra but not in the long-lived isotopes. We conclude that the enrichment at shallow station 404 (6m water depth) was due to resuspension by the strong winds. The coincidence of both short-lived and long-lived isotope enrichment at the bottom of stations 103 and Y points at SGD as the source. These observations support the interpretation that the high short-lived Ra activities observed at nearby station 1529 in winter 2015 were not due to resuspension but to SGD.

**3.4 Ra isotope constraints on the origin of water masses**

Figure 12a shows a theoretical mixing diagram; the corners represent the endmembers of the constituent water masses. Taken together, the data for the thermohaline structure, in combination with the high levels of activity of short-lived $^{224}$Ra and $^{223}$Ra in these plumes, which are much higher than in the river water and in bottom waters elsewhere, leads to the conclusion that this thermohaline anomaly is the result of SGD. This interpretation was also supported by summer data for long-lived Ra isotopes (section 3.3).

The data in Fig. 12 show the radium endmember concentrations and activity ratio in seawater (SW, bottom water), river water (RW, river plume) and transformed groundwater (TGW, the groundwater after mixing with seawater). We have separated out two types of water that form because of the mixing of these three sources with each other (Table 1, Fig. 12): brackish water (BW) and a mixture of ground water, river water, and seawater (MGRSW).

GW samples for end – member determination were obtained from sampling the two drilling wells, with the depth of the boreholes (the top of the bottom sediments is taken to be zero) being 18 and 15m. The lithological type of these sediments was water-bearing sands. The $^{224}$Ra/$^{223}$Ra activity ratio in the GW was 26.5. The $^{224}$Ra activity was in the range of 207 to 224dpm 100L$^{-1}$. GW is not represented on the graph due to a lack of salinity data.

The RW component is characterized by activities of $^{224}$Ra and $^{223}$Ra that are higher than in SW, but lower than in TGW (Table 1, Fig. 12). The $^{224}$Ra/$^{223}$Ra ratio represents the smallest values in comparison with the other endmembers.

The activities of $^{224}$Ra, $^{223}$Ra and the $^{224}$Ra/$^{223}$Ra ratio are almost unchanged throughout the SW mass. At the same time, the values of radium activity were among the lowest, comparable with the values of BW, while the $^{224}$Ra/$^{223}$Ra activity ratio had the highest values (Table 1, Fig. 12).

The TGW component is formed by mixing GW and bottom SW. These waters were discovered at the bottom of stations 1529, 1520, and 1504 in low-temperature fluid (Figs. 6, 7, 9). In addition, this water mass was detected at a depth of 6m at station 1507 (Fig. 9). This water mass is characterized by the highest values of radium activity (Table 1, Figs. 8, 9, 12).

The BW is mainly found in the surface horizon and forms as a result of mixing RW and SW. It has a salinity range of 3.8 to 11.9, with large dispersion of $^{224}$Ra activity (6.9 – 1.8dpm 100L$^{-1}$) and $^{224}$Ra/$^{223}$Ra ratio (4.3 – 36.4).

The points representing MGRSW lie in the central part of the mixing triangle (Fig. 12a) and in the middle of the 12b graphics, and are characterized by high or similar $^{224}$Ra and $^{223}$Ra activity in comparison with RW and SW. Moreover, the MGRSW salinity is less than in SW and TGW. The activity ratio of $^{224}$Ra/$^{223}$Ra varies from 18 to 29.

The data in Fig. 12 clearly localize three main and two additional water masses which result from the mixing of the main water mass types. Some boundary overlap is observed in the formed SW and TGW fields because short-lived isotopes can be modified by decay in addition to mixing (Fig. 12a). However, using additional parameters such as temperature, density, and activity ratio, as well as data on the thermohaline structure, we were able to separate these two signals (Table 1, Figs. 7, 9, 12b).

The close $^{224}Ra/^{223}Ra$ activity ratio of these two water masses (Table 1, Fig. 12b) is explained by the large participation of SW in TGW formation after GW discharge. As the result of this mixing, the highest activities of $^{224}Ra$ and $^{223}Ra$ were measured in this water; these isotopes are excellent tracers of SGD (Moore and Arnold, 1996; Rama and Moore, 1996; Charette et al., 2008). The $^{224}Ra/^{223}Ra$ ratio in the discharge location was 29, which also points to a mixing of GW and bottom SW (GW $^{224}Ra/^{223}Ra=26$; SW average of $^{224}Ra/^{223}Ra=35$). However, this is not a feature of SGD at the locations of the station 1504 and 1505 bottom horizons. First of all, the samples were taken 1m from the bottom (see methods) and it was precisely at these stations that the low-temperature fluid was closest to the bottom (Fig. 7); second, the $^{224}Ra/^{223}Ra$ ratio at these stations was less than at station 1529, with similar values of salinity, which shows $^{224}Ra$ decay with time after SGD. The other situation is characteristic of the bottom horizon at station 1520; here SGD was also not discovered by temperature. However, $^{224}Ra$ and $^{223}Ra$ activity and the $^{224}Ra/^{223}Ra$ ratio were comparable with those measured at station 1529 (Table 1, Fig. 8). Additionally, a shifting of isohalines was observed at station 1520 similar to that observed at station 1529 (Fig. 7), which points to mixing. This circumstance can be explained in two ways; either the source of the SGD is located close to station 1520, or the discharge has a pulsating character.

The $^{224}Ra/^{223}Ra$ activity ratio in the SW (bottom water) is higher than in the TGW, BW, RW and MGRSW. This $^{224}Ra/^{223}Ra$ activity ratio is possibly explained by formation of a high density near – bottom winter nepheloid layer caused by flocculation of humic substances and mineral particulates due to a lack of active mixing. Sinking Th isotopes scavenged from surface horizons of the water column accumulate on the organic and mineral particles in the bottom nepheloid horizon. Because of the much greater activity of $^{228}Th$ (half-life = 1.9 years) with respect to $^{227}Th$ (half-life = 18.9 days) the $^{227}Th$ decays faster than the $^{228}Th$; as a result, after 5 months of winter the $^{228}Th/^{227}Th$ and $^{224}Ra/^{223}Ra$ ratios increase.

The river endmember is well identified in the theoretical mixing diagram; some displacement of station 1531 seen in Fig. 12a is explained by the decay of $^{224}Ra$. This is indicated by a decrease in the $^{224}Ra/^{223}Ra$ activity ratio at this station.

We detected MGRSW in the surface plume at station 1520 ($^{224}Ra$ activity is 1.3 times higher here than at the surface of the neighboring stations 1529 and 1504), and in the waters on the periphery of the ice hummocks (called *stamukha* in Russian) in the shallows around Cape Muostakh. This MGRSW features higher $^{224}Ra$ activity compared to BW, as well as higher salinity and a higher $^{224}Ra/^{223}Ra$ activity ratio. In addition, station 1501 lies above the mixing line of the RW and SW endmembers, which also indicates the involvement of GW in the formation of this water mass.

Considering all the above data together, we have constructed a scheme for mixing water masses at the SGD site (Fig. 13). At the first stage, discharge GW is mixed with inflow bottom SW. After mixing, the TGW occupies an intermediate position between the bottom SW and surface RW-BW water according to its density. As a result, and because of the absence of wind mixing beneath the ice, a highly stratified water mass is formed (Fig.5). Due to the existence of a natural barrier in the form of the delta edge and the Lena River runoff, the distribution of TGW to the west is limited; it is found primarily east and south of the investigated area (Figs. 6, 7). The constant GW flow contributed to the formation of a small field of higher salinity water in comparison with the surrounding waters on the surface, slightly shifted to the east under the influence of Lena River runoff.  We also identified these waters as MGRSW (Fig.12, 13).

### 3.5 Estimate of SGD in the open talik zone

Estimate of SGD in the open talik zone calculated using Eq. (2) and from ex$^{224}Ra$ activities and the box (prism) model is represented in Fig. 14. *Vbox* is the volume of the box which is shaped like a prism, defined by corners with known

ex$^{224}$Ra activity in a TGW plume and with height equal to the average thickness of this plume. The average plume thickness was obtained from the density data for each TGW sample (Table 1, Fig.7). TGW density ranged between 1016.5 – 1018kg m$^{-3}$; thus, the average thickness (prism height) of the plume is 2.3m. The average ex$^{224}$Ra TGW activity was chosen as $A_{box}$. The sum of the average ex$^{224}$Ra SW and BW activities was chosen as $A_{offshore}$, because

these water masses have the maximum distribution in the Buor-Khaya Gulf study region, both horizontally and vertically. We did not use RW or MGRSW data in calculations, because their distribution is very local and therefore they can be neglected. The GW end – member obtained from sampling the two drilling wells represented $A_{GW}$. Residence time ($\tau$) was estimated using Eq. (1) for three different directions of TGW plume water transport from the place of SGD discharge (Fig. 14).

One of the main conditions for a radium age model (residence time) is the lack of radium sources after the first enrichment has occurred; the $^{224}$Ra/$^{223}$Ra activity ratio is initialized to a constant value and only changes by decay (Moore and de Oliveira, 2008; Dulaiova et al., 2006). In our case, under winter conditions and at shallow depths (4-10 m), there exist two primary water masses, the upper BW and the bottom SW; both are radium sources. This thermohaline structure is also complicated by the intrusions of pressurized GW discovered here. We note that the

presence of 2m thick fast ice excludes quick wind mixing, which to a certain extent simplifies the discovery of points for performing radium age modeling. The absence of significant BW, TGW, and bottom SW mixing is confirmed by the stratification factor data (Fig. 5) and the thermohaline structure (Fig. 7). Thus, the contribution of radium from the underlying BW layer and the overlying SW layer can be neglected. The TGW itself is well mixed, as indicated by the small range of thermohaline characteristics.

We took the value at bottom station 1529 to be the initial $^{224}$Ra/$^{223}$Ra activity ratio for calculating "radium age" in the fluid arriving as GW. According to all of the data presented above, it is precisely in this place that the SGD occurs. As is noted above, the $^{224}$Ra/$^{223}$Ra activity ratios of the GW from the boreholes and the bottom water differ, and apparently when they first arrive they mix. This fact disqualifies the use of the GW activity ratio from the borehole. In our case, we are dealing with a new water mass: TGW, the "lifetime" of which starts from the moment the bottom

SW mixes at station 1529.

The resulting radium ages were 3.2, 1.9, and 1.5 days for 1504 (bottom), 1505 (bottom), and 1507 (6m), respectively. The values obtained show that the radium age to the south (1.5 days) and southwest (1.9 days) of the TGW plume is less than to the northwest (3.2 days), despite the fact that the distance here is much greater (Fig. 14). This result is completely logical, because our TGW plume meets a river plume from the northwest; as a result, the TGW flow rate

decreases in this direction. Furthermore, the bottom relief has a certain impact. The bottom has drop-offs in depth (delta edge), which coincide with the western boundaries of the fluid; therefore, the speed of fluid distribution in the west is significantly lower than in the south and the east (Figs. 2, 6, 7). The thermohaline structure also indicates this. It is clearly visible that the place of discharge (station 1529) has shifted to the northwest with respect to the body of the fluid (Figs. 6, 7).

We measured the current speed in the fluid (6m depth) at oceanographic station 1507 with an Acoustic Doppler Current Profiler (ADCP). The residual current speed was 3.8cm s$^{-1}$; it traveled 3.3km d$^{-1}$. From radium data at station 1505, we derived a radium age of about 2 days. The distance from the SGD location (bottom of station 1529) was around 7km, indicating that the TGW distribution speed was equal to 3.5km d$^{-1}$, in perfect agreement with the independent ADCP measurement of the current.

Taking together all the above assumptions and calculations, we obtained an SGD discharge equal to $1.7 \times 10^6$m$^3$ d$^{-1}$ or 19.9m$^3$ s$^{-1}$. These values are insignificant if compared with the average Lena River April discharge (1426m$^3$ s$^{-1}$) (http://rims.unh.edu/data.shtml), but if we compare the obtained SGD discharge data with the April Yana River

discharge ($0.68m^3$ $s^{-1}$ [http://rims.unh.edu/data.shtml]), the SGD contribution to the terrestrial water budget in the study area is more than one order of magnitude greater. Keeping in mind that that the SGD values were evaluated for the small area of Buor Khaya Gulf which is only a tiny fraction of the ESAS, the total SGD signal for the entire ESAS must be orders of magnitude higher.

### 3.6 Mechanisms of submarine groundwater discharge

### 3.6.1 Discharge of the subpermafrost water

The study area is located in a permafrost region, which complicates not only discharge, but also GW recharge. Therefore, in our case, the only means of recharging is via open taliks. Such terrain features facilitate interaction of subpermafrost water with the suprapermafrost water which forms in the valleys of large rivers, by lakes, and in fault zones. In the river watershed, the suprapermafrost water collects from surrounding slopes (Fig.15a) (Pinneker, 1983, Romanovskii, 1983). In the current study area, one of the main sources of GW recharge may be the large Lena River,

which flows several hundred km from our SGD, along the other side of the Kharaulakh Ridge (Fig. 15b).

It is known that open taliks exist in the Lena River bed (Antonov, 1987; Semiletov et al., 2000) in the area of the West-Verkhoyan Fault (Fig. 15b), hydraulically connected by fissures and faults with the GW of the entire hydrogeological massif (Fig. 15a). Additionally, researchers have noted that the level of water discharge arriving in the Lena River Delta is higher than that measured at the outlet of the river (Bolshiyanov et al., 2013; Fedorova et al.,

2015). This difference in elevation is explained by the complex hydrographic layout of the delta and peculiarities of delta geological and geomorphological structure, and it creates a pressure gradient for submarine GW discharge.

The waters of the open taliks of large rivers and lakes penetrate through the permafrost and mix with the cryogenic GW. This water formed as the result of multi-year freezing of rocks and the simultaneous concentration of salts (Pinneker, 1983; Romanovskii, 1983), resulting in total dissolved solids in the cryogenic GW from $1 \times 10^4$ to $30 \times 10^4$

25      mg/L and a temperature from 0 to -12°C. In the Kharaulakh hydrogeological massif, the thickness of the cryogenic groundwater reaches 300-400m; the permafrost is 500-700m thick (Romanovskii, 1983) (Fig. 15a). In the end, the cryogenic GW contributes to the cold and salty GW which is released as SGD, and after mixing with bottom SW, transforms into the cold (about -1°C) and saline (salinity 22) TGW.

The study area is part of the Kharaulakh hydrogeological massif (north of the Verkhoyansk Fold) which is complicated

by listric faults along the coastline (Romanovskii, 1983; Imaev et al., 2004) (Figs. 3, 4). The permeability and water-bearing capacity of the lithified rocks of the hydrogeological massif is governed by the presence of fissures, faults, and pores (Pinneker, 1983; Romanovskiy, 1983). Usually the young active faults are the most crushed and broken (Sherard et al., 1974); therefore, in our case, the Neogene-Quaternary faults along the Buor-Khaya Gulf coastline have favorable conditions for the transport of pressurized subpermafrost water. The lack of permafrost over the fault zones

removes the last obstacle to the path of flowing GW before it is discharged into the Buor-Khaya Gulf water column. The small shift in SGD location to the east relative to the fault line is possibly caused by the presence of a fine sediments stratum (a confining bed), which we discovered by TEM on top of the listric fault footwall (Fig. 15c). The fact that the summer and winter SGD springs were found on a line parallel to the fault once again points to the connection between the tectonogenic talik and the SGD (Fig. 4).

The SGD is likely located in this sector of the fault precisely because here the listric fault crosses another fault with unknown kinematics (Fig. 4a). It is known that where two faults of different orientations meet or cross each other a large part of the rock masses suffers from increased crushing or jointing (Selmer-Olsen, 1964), and this in turn creates

favorable conditions for rising GW. The region west of our SGD sites has a similar tectonic structure (fault crossing). However, there are no SGD features, which may be explained by the following. The surface of this area is categorized as vast shoals; for the greater part of the year it is covered by fast ice and freezing ice hummocks containing sediments. Furthermore, this area was flooded much later by the transgression of the sea, and possibly these shoals arose exclusively due to thermal abrasion after the time of marine transgression, which gives us grounds to hypothesize that the permafrost starts at shallow depths below the surface sediments and thus hinders SGD (Fig. 15a).

**3.6.2 Cryogenic squeezing out of brine (CSB) and expulsion of water-soluble salts**

Besides the fluid and river plume examined above, there was one more place in the studied area which also had high activities of the isotopes $^{224}$Ra and $^{223}$Ra. The radium activities were significantly higher than in the river plume on the periphery of the ice hummocks at the Cape Muostakh shoals (station 1501). Moreover, the salinity of the entire column exceeded 12 (Table 1, Fig. 6), and furthermore, this point was located in the region impacted by Lena River drainage (Charkin et. al., 2011) and far from our SGD which was under pressure. During our winter studies, the air temperature did not rise above -10°C during the day, and at night it dropped to -30°C. The temperature of the water was negative everywhere. On the graph of $^{224}$Ra and $^{224}$Ra/$^{223}$Ra dependency on salinity (Fig. 12), this point lies in the region of the MGRSW. Taken together, these findings indicate that this phenomenon is likely the result of CSB and expulsion of water-soluble salts from the ice formation zone into the warmer unfrozen parts of the sediments due to a compression and crystallization effect. This phenomenon has been described as permafrost on land (Tyutyunov, 1966; Baker and Osterkamp, 1989; Shepelev and Sannikova, 2001), and we may have noted it for the first time here in the sea bottom sediments.

During the autumn-winter formation of the fast ice, huge areas of the Buor-Khaya Gulf shoals freeze. As the result of the freezing of water-bearing sands, CSB and expulsion of water-soluble salts occurs in the part which has not frozen. Figure 16 shows the mechanism of this SGD formation. Under the impact of a hydraulic piston, the water is confined by the bed of clay (documented in drill borehole 4D-12 [Figs. 4a, b]), which is located together with a cryogenically confining bed of submarine permafrost. As a result, the brine flows along the path of least resistance, enters the water column of the adjacent water.

Despite the large area of the Buor-Khaya Gulf shallows and bars, we have not detected a strong influence of this process on the thermohaline structure of the investigated area. This phenomenon is observed at the place of direct contact between an ice hummock and the bottom sediments. The coastal orography also is an important factor. The location of the discovered phenomenon is isolated from the main RW flow by a spit of land which restricts removal of discharged GW (Figs. 2, 4, 6).

**4 Conclusions**

The features and nature of subpermafrost GW discharge in the Siberian Arctic seas depend on the thermal state of the permafrost as well as on the geological and tectonic structure of the shelf. The geological prerequisites for subpermafrost GW discharge include the presence of lithological conditions (sands, gravel, cracks and fissures in rocks) and channels (taliks) between the subpermafrost GW (confined aquifer) and the marine water column. From the tectonic position point of view, the most favorable conditions for the discharge of pressurized subpermafrost GW are formed in the rifting of active fault zones, especially at the locations of fault crossings. At fault crossings, first,

there is an increased crushing or jointing of rock masses, which is favorable for uprising GW transport, and second, the impact of geothermal heat flux is increased, which thaws the permafrost.

We obtained a submarine groundwater discharge equal to $1.7 \times 10^6 m^3 d^{-1}$ or $19.9 m^3 s^{-1}$. These values are insignificant if compared with the average Lena River April discharge ($1426 m^3 s^{-1}$), but if we compare the obtained SGD discharge

data with the April Yana River discharge ($0.68 m^3 s^{-1}$), the SGD contribution terrestrial water budget in the study area is more than one order of magnitude greater. Keeping in mind that that the SGD values were evaluated for the small area of Buor Khaya Gulf which is only a tiny fraction of the ESAS, the total SGD signal for the entire ESAS must be orders of magnitude higher.

Another mechanism of SGD are being formed in the bottom sediments of shoals, newly frozen sediments function as

a hydraulic piston pushing interstitial brine and salt into the water column. However, we consider such GW production to be less significant than the mechanism of pressurized GW contributing to the overall SGD.

This study is the first to directly observe submarine GW discharge along the Eurasian Arctic margin. It introduces a mechanistic context for this process. Future studies should seek to expand on the multi-proxy approach to constrain the quantitative significance of this transport vector, not only for terrestrial discharge but also for associated old carbon

and methane release from the permafrost system.

**Data availability**

All data are available in Table 1, 2 as well as in Supplementary Table S1.

*Acknowledgements* This work was supported by the Russian Government (mega-grant under contract No. 14.Z50.31.0012). Financial support has also been provided by the US National Science Foundation (Nos. OPP ARC-1023281; 0909546) and the NOAA OAR Climate Program office (NA08OAR4600758). A.Ch., N.S., and O.D. acknowledge support from the Russian Science Foundation (grant No. 15-17-20032). O.G. thanks the Swedish Research Council (VR), the Knut and Alice Wallenberg Foundation, and the European Research Council (ERC-AdG

project CC-TOP#695331 to O.G.). MRvdL thanks captain and crew of the R/V *Dalnie Zelentsy* for their help during field sampling, and Waldemar Schneider and Alexandra Kraberg for their support in organizing the Lena Delta 2013 expedition. We thank Dmitry Melnichenko and Tiksi Hydrobase for logistical support with the multi-year winter oceanographic/drilling campaigns (2011-2015). We thank Candace O'Connor for English editing.

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

**Table 1. Salinity, temperature (°C), density (kg m$^{-3}$), activities of short-lived radium ($^{224}$Ra and $^{223}$Ra) isotopes (dpm 100L$^{-1}$), and $^{224}$Ra/$^{223}$Ra activity ratio at the sampling sites in the western Buor-Khaya Gulf (Laptev Sea) during winter.  GW: ground water, TGW: transformed ground water, RW: river water, BW: brackish water, SW: sea water, MRGSW: mixture of GW, RW, SW.**

| Station ID | Depth, m | Water type | Salinity | Temperature | Density | ex $^{224}$Ra | $^{223}$Ra | 224/223 AR |
|---|---|---|---|---|---|---|---|---|
| **ID15** | 18 m | GW | -* | -* | -* | 207 ± 3.9 | 7.8 ± 0.65 | 26.5 ± 1.8 |
| **IID14** | 15 m | GW | - | - | - | 224 ± 2.5 | 8.5 ± 0.48 | 26.4 ± 0.9 |
| **GW average (n=2)** | | | - | - | - | 215 ± 3.2 | 8.15 ± 0.56 | 26.4 ± 1.3 |
| **Standard deviation** | | | - | - | - | 12 | 0.49 | 0.07 |
| **1529** | bottom | TGW | 22.1 | - 1.01 | 1017.8 | 19 ± 0.44 | 0.65 ± 0.06 | 29.2 ± 1.7 |
| **1520** | bottom | TGW | 22.2 | - 0.96 | 1017.8 | 16.8 ± 0.55 | 0.58 ± 0.07 | 29 ± 2.2 |
| **1504** | bottom | TGW | 22.4 | - 1 | 1018 | 14.5 ± 0.55 | 0.75 ± 0.10 | 19.3 ± 1.6 |
| **1505** | bottom | TGW | 22.1 | - 0.97 | 1017.7 | 14.3 ± 0.48 | 0.62 ± 0.08 | 23.1 ± 1.9 |
| **1507** | 6 m | TGW | 20.9 | - 0.98 | 1016.5 | 8.2 ± 0.25 | 0.34 ± 0.04 | 24.1 ± 1.8 |
| **TGW average (n=5)** | | | 22 | -0.98 | 1017.6 | 14.6 ± 0.45 | 0.59 ± 0.08 | 24.9 ± 1.8 |
| **Standard deviation** | | | 0.66 | 0.02 | 0.6 | 4.04 | 0.15 | 4.2 |
| **1501** | mid | MRGSW | 12.4 | - 0.57 | 1010 | 10.9 ± 0.39 | 0.44 ± 0.06 | 24.7 ± 2.1 |
| **1520** | surface | MRGSW | 11.5 | - 0.67 | 1009.2 | 7.6 ± 0.38 | 0.42 ± 0.07 | 18 ± 1.9 |
| **1508** | surface | MRGSW | 10.9 | - 0.58 | 1008.7 | 7.3 ± 0.34 | 0.25 ± 0.05 | 29.2 ± 3.6 |
| **1529** | 6 m | MRGSW | 17.8 | - 1 | 1014.3 | 5.4 ± 0.34 | 0.21 ± 0.04 | 26.1 ± 3.3 |
| **MRGSW average (n=4)** | | | 13.1 | -0.71 | 1010.5 | 7.8 ± 0.36 | 0.33 ± 0.05 | 24.5 ± 2.7 |
| **Standard deviation** | | | 3.16 | 0.2 | 2.5 | 2.28 | 0.12 | 4.72 |
| **1502** | mid | RW | 0.98 | - 0.06 | 1000.6 | 9 ± 0.37 | 0.55 ± 0.07 | 16.4 ± 1.4 |
| **1515** | surface | RW | 1.6 | - 0.09 | 1001.2 | 6.3 ± 0.35 | 0.4 ± 0.07 | 15.7 ± 1.8 |
| **1531** | mid | RW | 1.73 | - 0.08 | 1001.3 | 1.8 ± 0.37 | 0.42 ± 0.08 | 4.3 ± 0.8 |
| **RW average (n=3)** | | | 1.4 | -0.07 | 1001 | 5.7 ± 0.36 | 0.45 ± 0.07 | 12.1 ± 1.3 |
| **Standard deviation** | | | 0.4 | 0.02 | 0.38 | 3.6 | 0.08 | 6.8 |
| **1530** | surface | BW | 3.8 | - 0.2 | 1002.9 | 4.3 ± 0.35 | 0.15 ± 0.05 | 28.6 ± 5.9 |
| **1503** | mid | BW | 4.1 | - 0.22 | 1003.2 | 5.1 ± 0.45 | 0.14 ± 0.04 | 36.4 ± 6.8 |
| **1532** | mid | BW | 6.7 | - 0.2 | 1005.3 | 4.7 ± 0.9 | 0.28 ± 0.11 | 16.7 ± 4.8 |
| **1514** | surface | BW | 7.1 | - 0.39 | 1005.6 | 5.2 ± 0.42 | 0.19 ± 0.05 | 27.3 ± 4.6 |
| **1507** | surface | BW | 7.8 | - 0.43 | 1006.2 | 5.9 ± 0.36 | 0.26 ± 0.05 | 22.7 ± 2.8 |
| **1506** | mid | BW | 8.5 | - 0.4 | 1006.8 | 5.6 ± 0.24 | 0.44 ± 0.06 | 12.7 ± 1.1 |
| **1504** | surface | BW | 9.4 | - 0.53 | 1007.5 | 5.7 ± 0.42 | 0.22 ± 0.05 | 25.9 ± 3.9 |
| **1509** | mid | BW | 9.6 | - 0.51 | 1007.6 | 6.9 ± 0.32 | 0.28 ± 0.05 | 24.6 ± 2.7 |
| **1505** | surface | BW | 9.8 | - 0.54 | 1007.8 | 6.2 ± 0.31 | 0.41 ± 0.07 | 15.1 ± 1.6 |
| **1529** | surface | BW | 11.9 | - 0.68 | 1009.5 | 5.7 ± 0.27 | 0.25 ± 0.04 | 22.8 ± 2.3 |
| **BW average (n=10)** | | | 7.9 | -0.41 | 1006.2 | 6 ± 0.4 | 0.26 ± 0.06 | 23.3 ± 3.6 |
| **Standard deviation** | | | 2.55 | 0.16 | 2.1 | 0.75 | 0.1 | 7.05 |
| **1507** | bottom | SW | 22.5 | - 0.9 | 1018.1 | 6.2 ± 0.24 | 0.19 ± 0.03 | 32.6 ± 3.2 |
| **1508** | bottom | SW | 22.4 | - 0.87 | 1018.1 | 7.2 ± 0.26 | 0.19 ± 0.02 | 37.8 ± 2.6 |
| **1530** | bottom | SW | 21.9 | - 0.94 | 1017.7 | 5.1 ± 0.37 | 0.14 ± 0.03 | 36.4 ± 5.1 |
| **1507** | 7 m | SW | 21.5 | - 0.97 | 1017.4 | 5.4 ± 0.22 | 0.16 ± 0.03 | 33.7 ± 3.8 |
| **SW average (n=4)** | | | 22.1 | -0.92 | 1017.8 | 5.9 ± 0.27 | 0.17 ± 0.02 | 35.1 ± 3.6 |
| **Standard deviation** | | | 0.46 | 0.04 | 0.34 | 0.94 | 0.02 | 2.39 |

*- measurements were not carried out

**Table 2. Salinity, activities of radium isotopes ($^{224}$Ra, $^{226}$Ra, $^{228}$Ra, dpm 100L$^{-1}$) in the wider Delta area (Laptev Sea) during the summer survey.**

| Station ID | Bot. Depth, m | Depth, m | Salinity | ex $^{224}$Ra | $^{226}$Ra | $^{228}$Ra |
|---|---|---|---|---|---|---|
| 101 | 4 | 0 | 0.8 | 17.5 ± 0.9 | 10.4 ± 0.6 | 16.4 ± 1.8 |
| 103 | 12 | 0.5 | 3.4 | 11.3 ± 0.7 | 7.1 ± 0.5 | 13.2 ± 1.5 |
| 103 | 12 | 6 | 4.5 | 7.9 ± 0.6 | 6.6 ± 0.5 | 12.2 ± 1.4 |
| 103 | 12 | 12 | 15.9 | 92.2 ± 1.9 | 31.5 ± 1.1 | 77.0 ± 4 |
| 105 | 18 | 2 | 7.1 | 7.4 ± 0.5 | 7.8 ± 0.5 | 18.5 ± 1.5 |
| 105 | 18 | 10 | 16.3 | 4.7 ± 0.5 | 12.8 ± 0.6 | 30.4 ± 1.9 |
| 105 | 18 | 17 | 26.3 | 16.1 ± 0.8 | 14.4 ± 0.7 | 40 ± 2.5 |
| 107 | 19 | 4 | 11.6 | 3.3 ± 0.4 | 11.4 ± 0.6 | 22.3 ± 1.7 |
| 107 | 19 | 11 | 17.2 | 3.7 ± 0.4 | 13.3 ± 0.6 | 32.5 ± 2 |
| 107 | 19 | 18 | 25.7 | 14.1 ± 0.8 | 13.6 ± 0.6 | 38.8 ± 2.3 |
| 401 | 20 | 4 | 4.5 | 7.8 ± 0.6 | - | - |
| 401 | 20 | 7 | 9.2 | 3.6 ± 0.5 | 7.7 ± 0.4 | 17.2 ± 1.3 |
| 401 | 20 | 16 | 20.6 | 6.8 ± 0.5 | 14.1 ± 0.6 | 35.2 ± 2.1 |
| 403 | 21 | 2 | 5.9 | 7.0 ± 0.6 | 9.7 ± 0.5 | 18.8 ± 1.5 |
| 403 | 21 | 8.5 | 19.3 | 8.9 ± 0.6 | 15.6 ± 0.6 | 34 ± 2.1 |
| 403 | 21 | 14 | 25.6 | 6.4 ± 0.5 | - | - |
| 403 | 21 | 20 | 27.2 | 9.6 ± 0.6 | - | - |
| 404 | 6 | 0.5 | 2.8 | 17.1 ± 0.9 | 10.7 ± 0.5 | 19 ± 1.5 |
| 404 | 6 | 2.5 | 21.8 | 43.4 ± 1.1 | 14.9 ± 0.5 | 37.2 ± 2 |
| 404 | 6 | 5 | 25.2 | 36.4 ± 1.2 | 14.5 ± 0.6 | 39.6 ± 2.3 |
| 501 | 25 | 2 | 4.7 | 6.1 ± 0.6 | 5.9 ± 0.4 | 13.5 ± 1.3 |
| 501 | 25 | 23 | 29.2 | 7.6 ± 0.7 | 13.4 ± 0.6 | 35.5 ± 2.1 |
| 503 | 16 | 4 | 3.9 | 5.3 ± 0.4 | 5.2 ± 0.4 | 13.5 ± 1.4 |
| 503 | 18 | 9.5 | 18.4 | 4.2 ± 0.4 | 15.2 ± 0.7 | 40.4 ± 2.4 |
| 503 | 18 | 14 | 24.8 | 8.2 ± 0.5 | 14.6 ± 0.6 | 42.8 ± 2.4 |
| 503 | 18 | 17 | 29.3 | 12.6 ± 0.8 | 14.4 ± 0.6 | 41.3 ± 2.4 |
| 504 | 6 | 1 | 17.4 | 25.3 ± 1.1 | 13.1 ± 0.5 | 32.8 ± 1.9 |
| 504 | 6 | 4.5 | 25.9 | 23.1 ± 1 | 14.1 ± 0.6 | 38.6 ± 2.2 |
| 601 | 27 | 3 | 16.4 | 5.6 ± 0.5 | 12.4 ± 0.6 | 30 ± 1.9 |
| 601 | 27 | 10 | 22.6 | 0.7 ± 0.3 | 10.4 ± 0.5 | 25.7 ± 1.8 |
| 601 | 27 | 23 | 30.3 | 9.2 ± 0.7 | 12.2 ± 0.6 | 34.3 ± 2.1 |
| 603 | 10 | 1.5 | 25.0 | 9.5 ± 0.6 | 14.8 ± 0.6 | 45.6 ± 2.5 |
| 603 | 10 | 3.2 | 25.5 | 8.7 ± 0.6 | 12.9 ± 0.6 | 43.4 ± 2.5 |
| 603 | 10 | 9 | 29.0 | 11.5 ± 0.6 | 13.9 ± 0.7 | 45.8 ± 2.7 |
| 604 | 2 | 0 | 2.6 | 15.7 ± 1.1 | 8.6 ± 0.6 | 17.9 ± 1.7 |
| 605 | 1.5 | 0 | 0.0 | 1.1 ± 0.4 | 2.3 ± 0.4 | 0.7 ± 1.1 |
| X | 6 | 0.5 | 8.3 | 21.4 ± 6.5 | 13.3 ± 0.6 | 27.9 ± 1.9 |
| X | 6 | 2.3 | 24.0 | 21.7 ± 5.9 | 14.8 ± 0.6 | 38.1 ± 2.3 |
| X | 6 | 5 | 25.1 | 16.0 ± 0.9 | 12.6 ± 0.6 | 36.6 ± 2.1 |
| Y | 12 | 1 | 3.2 | 10.0 ± 0.6 | 9.6 ± 0.5 | 15.8 ± 1.4 |
| Y | 12 | 10 | 21.8 | 69.1 ± 1.3 | 27.8 ± 0.9 | 75.7 ± 3.8 |

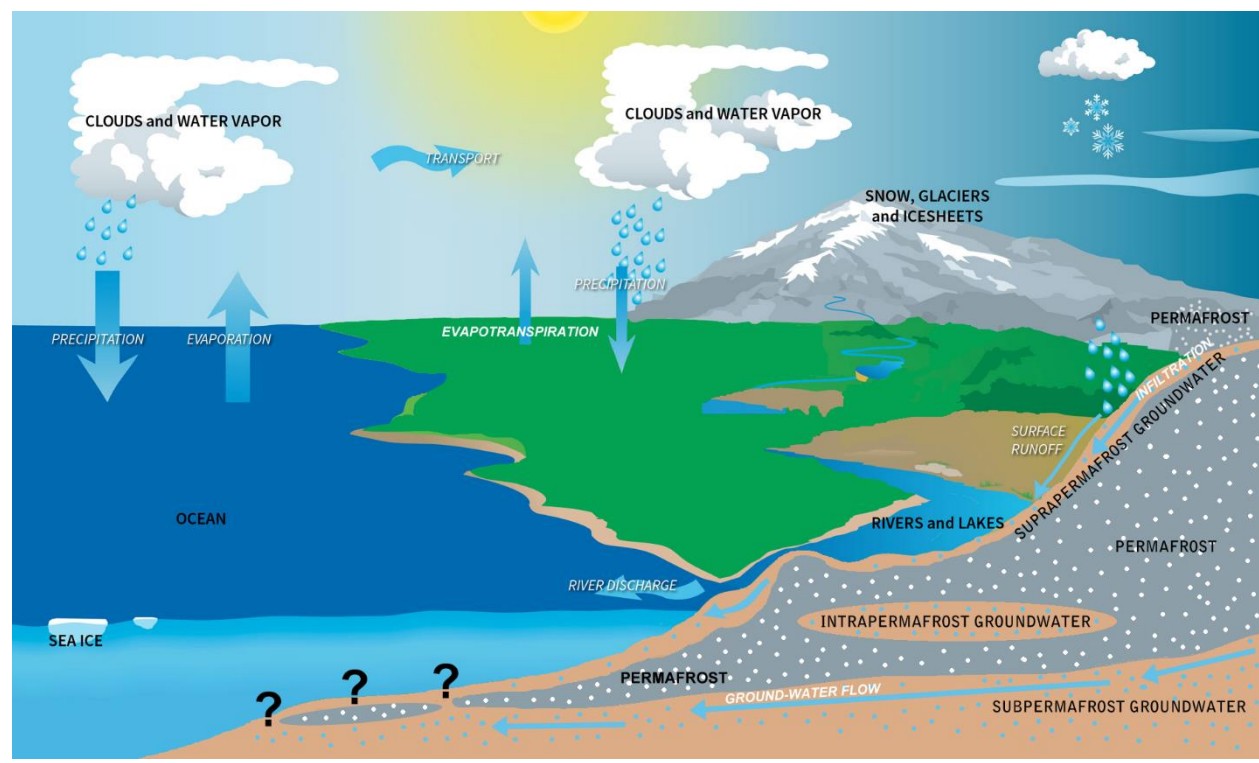

**Figure 1: A view of the Arctic hydrological cycle showing key linkages among land, ocean, and atmosphere (modified from http://arcticchamp.sr.unh.edu/whatisarctichydro.shtml).**

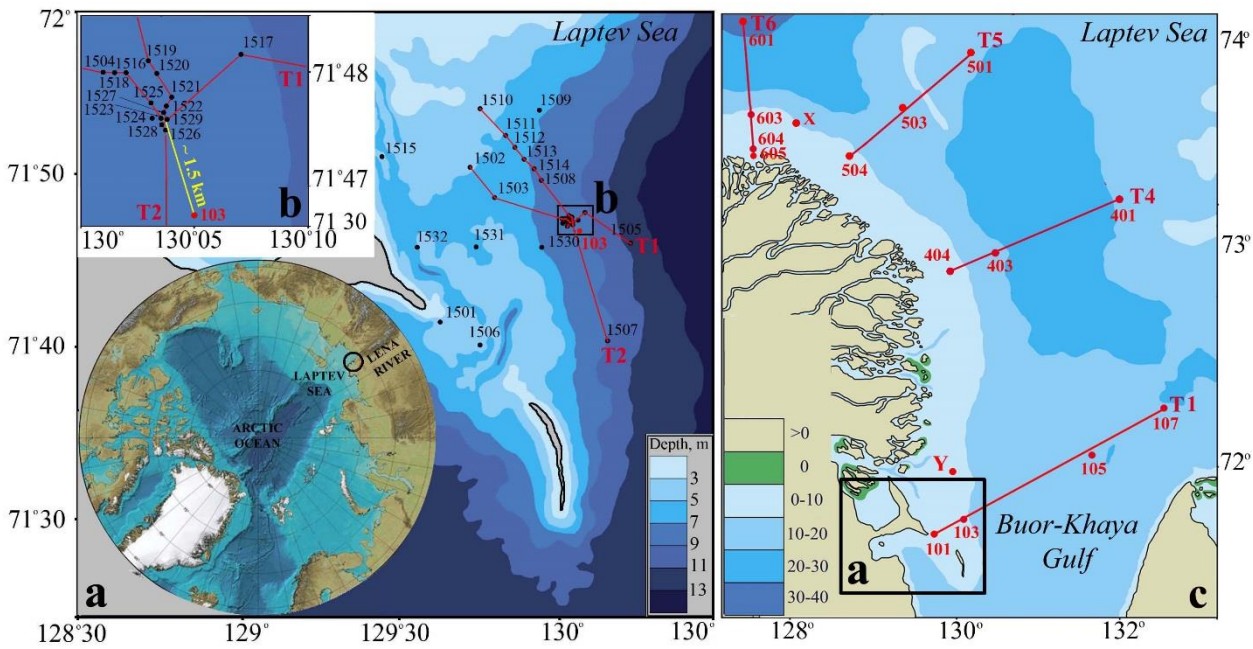

**Figure 2: Study area: a) Study area in winter, b) detail of winter observation sites. Yellow line and number, the approximate distance between winter (no. 1529) and summer (no. 103) stations, c) study area in summer. Black dots – location and numbers of winter oceanographic stations, red dots - location and numbers of summer oceanographic stations, red lines - oceanographic transects of temperature, salinity and their numbers. Background map modified from IBCAO (www.ngdc.noaa.gov/mgg/bathymetry/arctic/arctic.html).**

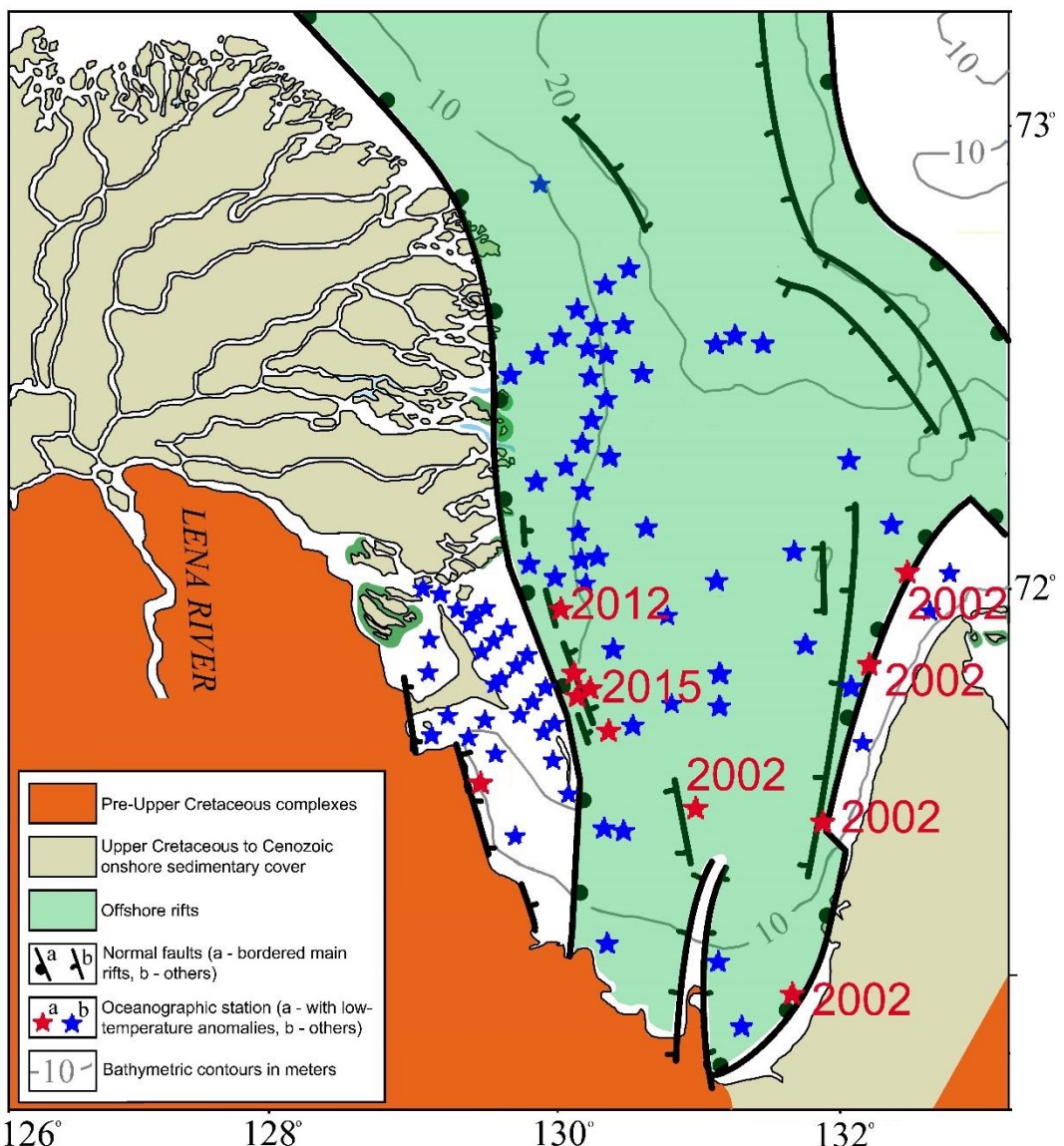

**Figure 3: Ust Lena graben (after Drachev et al., 1998; Franke et al., 2001; Imaev et al., 2004). Numbers above the red stars indicate the years when the studies were carried out.**

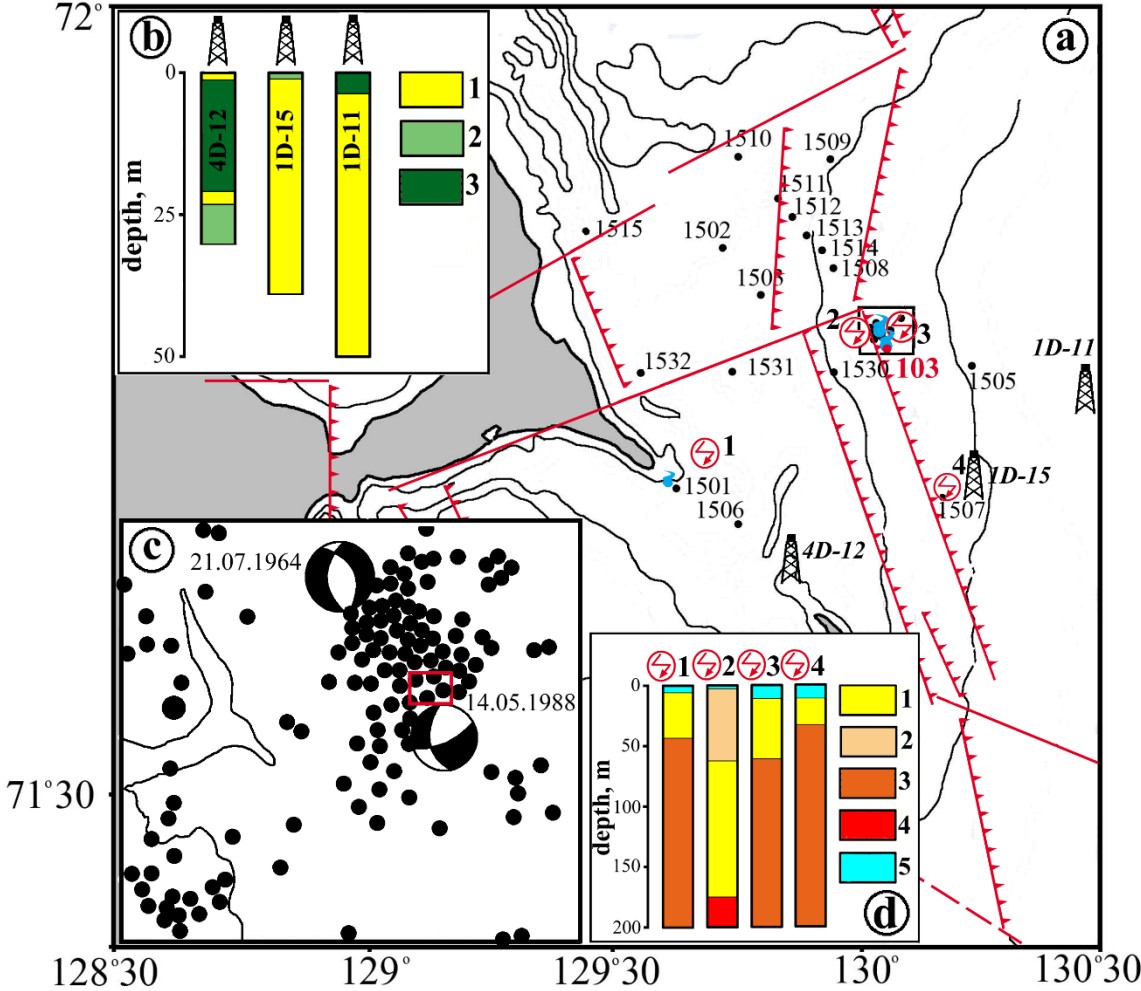

**Figure 4: Location of drilling sites and oceanographic stations superimposed on a seismotectonic map modified from Imaev et al. (2004) and overview map of Fig. 2.**

a) The red lines with teeth are active faults; the red lines are active faults with kinematics, which have not yet been established; the red dashed lines are the presumed faults; the red rings are the geoelectric survey locations; the blue dots with a tail are the locations where SGD was discovered; the black tower are the locations of drilling operations.

b) Grain size composition of the drilling cores 1D-11, 1D-15, 4D-12. 1) Sands, 2) predominantly silts, 3) predominantly clays.

c) The map of the earthquake epicenters and focal mechanisms. The small dots are earthquakes with a magnitude ≤ six. The large circles are earthquakes with a magnitude > six, placed near the date of their occurrence.

d) Resistivity in the bottom sediments (Ω m): 1) 0-5, 2) 5-10, 3) 35-100, 4) >350, 5) water. A table with more detailed information can be found in Supplementary Table 1.

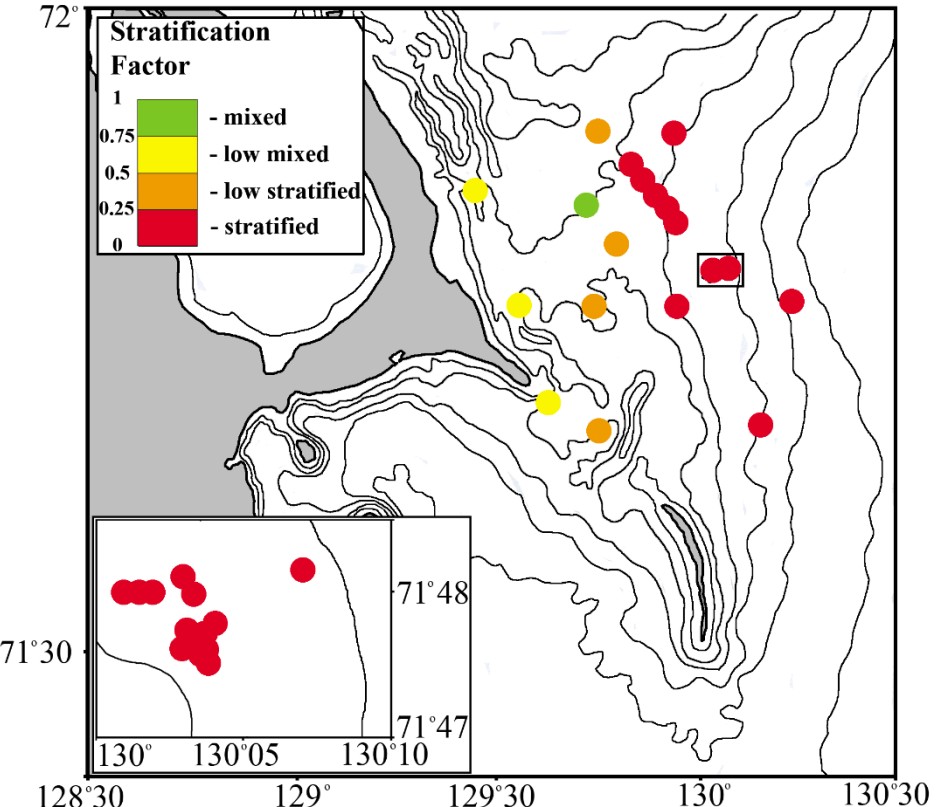

**Figure 5: Winter stratification factor in the study areas.**

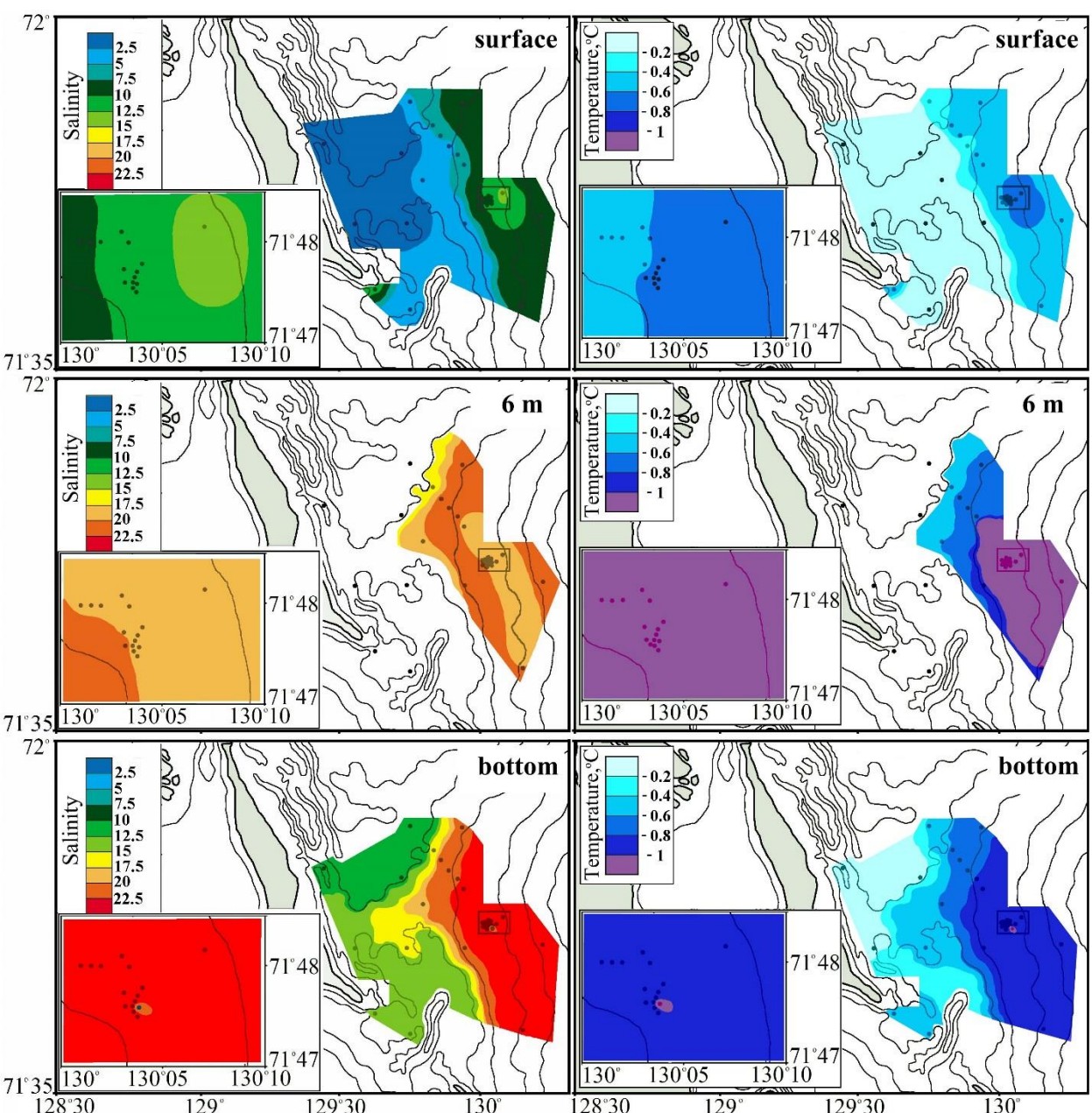

**Figure 6: Salinity and temperature distribution of water at the surface, 6 m deep, and at the bottom in the winter.**

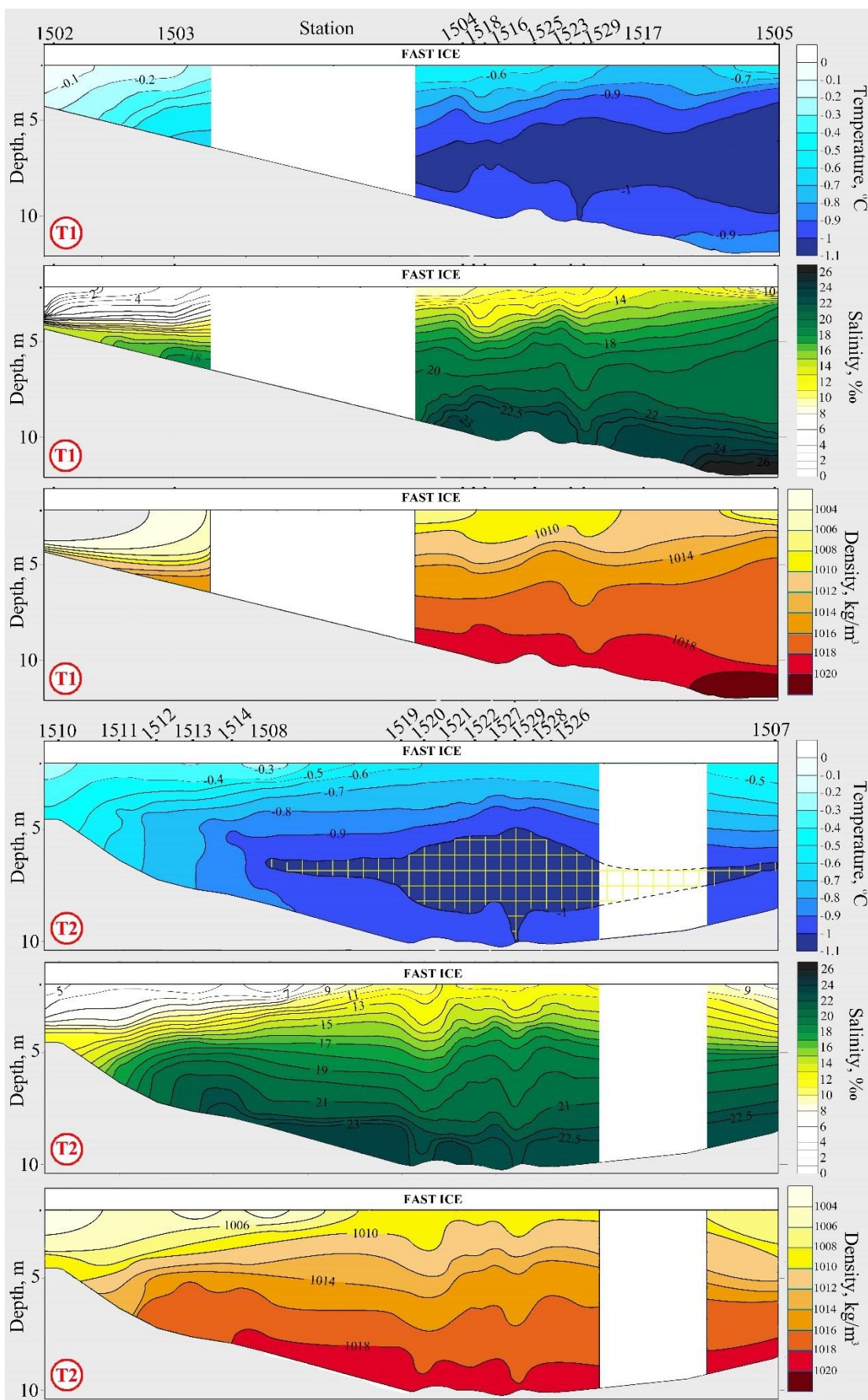

**Figure 7: Vertical sections of winter salinity, temperature, and density distributions. The location of transects T1 and T2 is shown in Figure 2. A yellow cell shows a "mushroom-like structure". In places where there are no data, white gap are shown.**

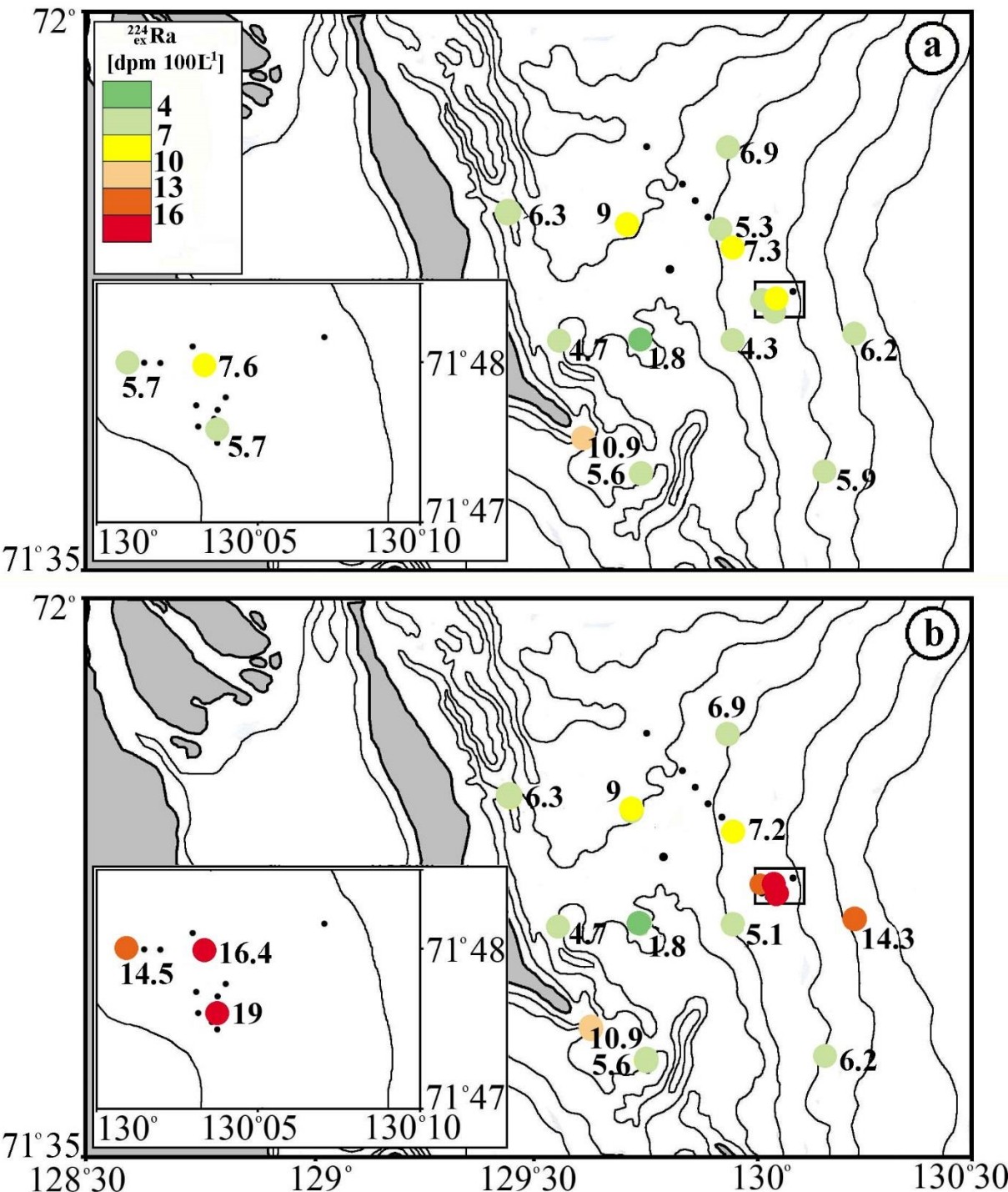

**Figure 8: Winter distributions of ex²²⁴Ra in surface (a) and bottom (b) water. At the shallow stations one value is shown for both horizons.**

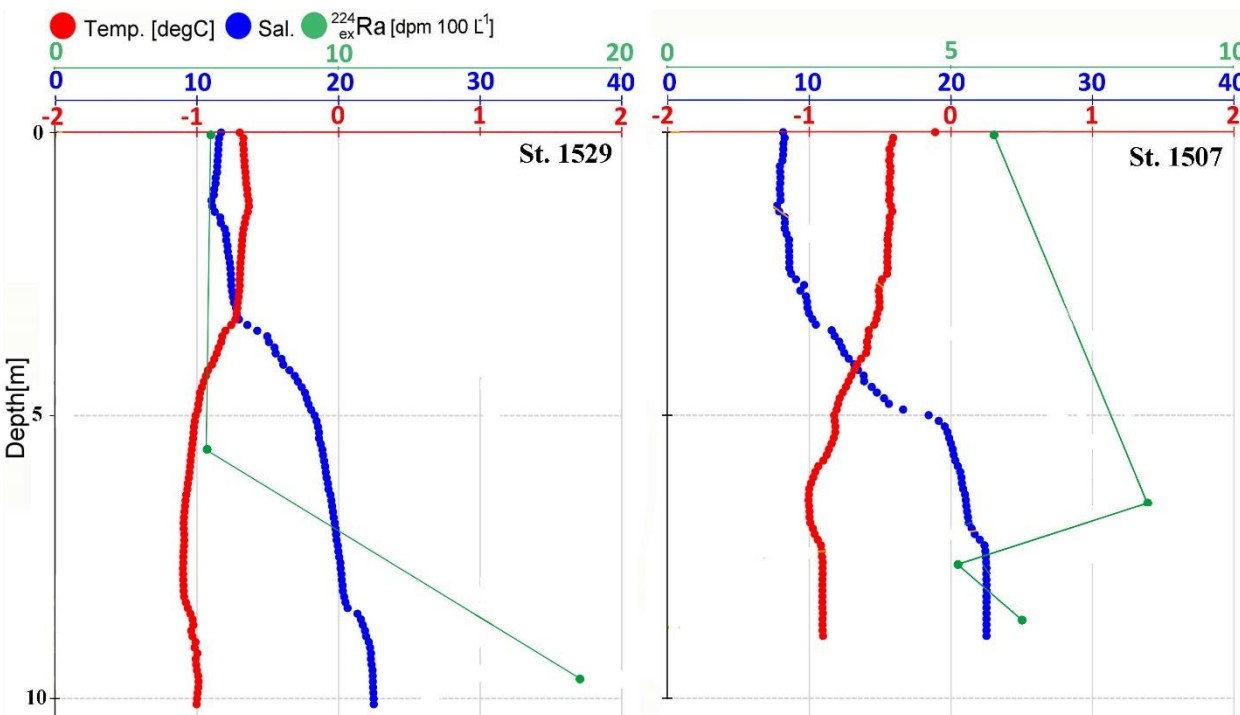

**Figure 9: Vertical profiles of the winter water column.**

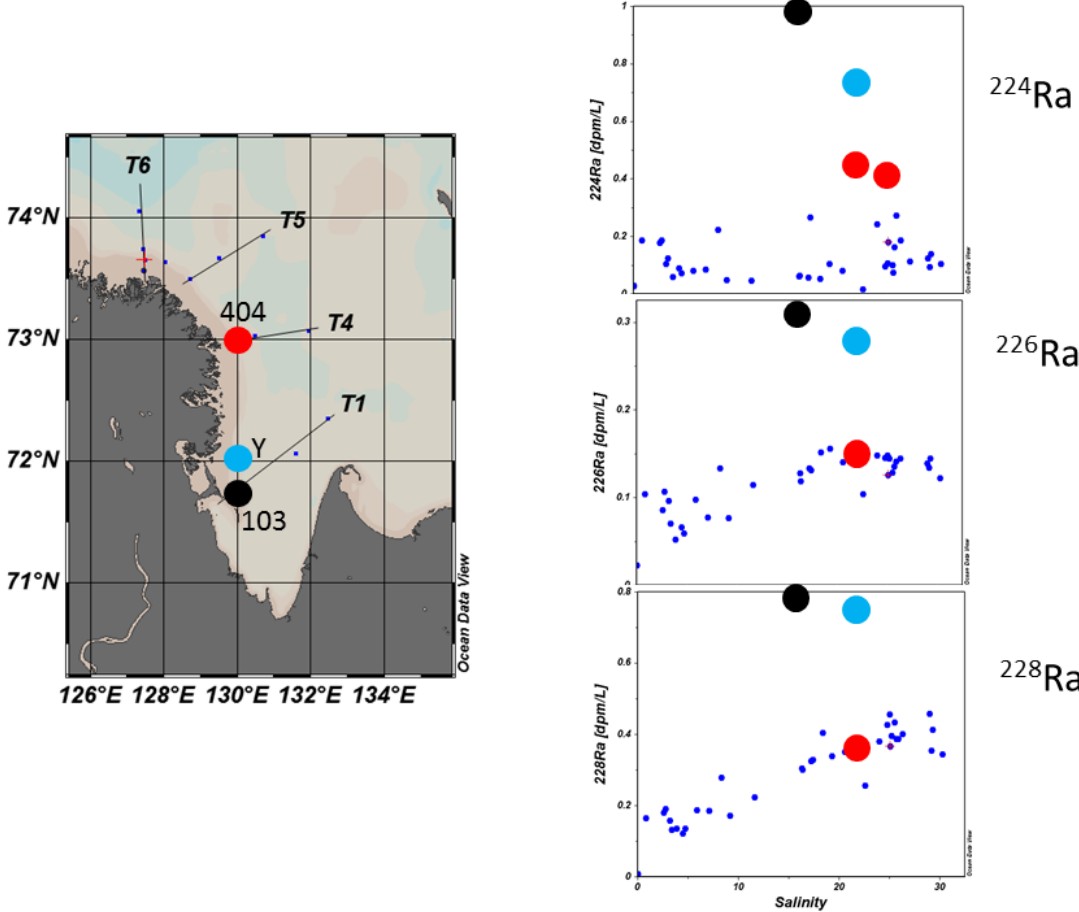

**Figure 10: Activities of ²²⁴Ra, ²²⁶Ra, and ²²⁸Ra as function of salinity during the summertime cruise. Bottom water activities at station 103 (12 m, black) and Y (10 m, blue) are the only ones that stand out for all three isotopes. Figure made with Ocean Data View (Schlitzer, R., Ocean Data View, http://odv.awi.de, 2017)**

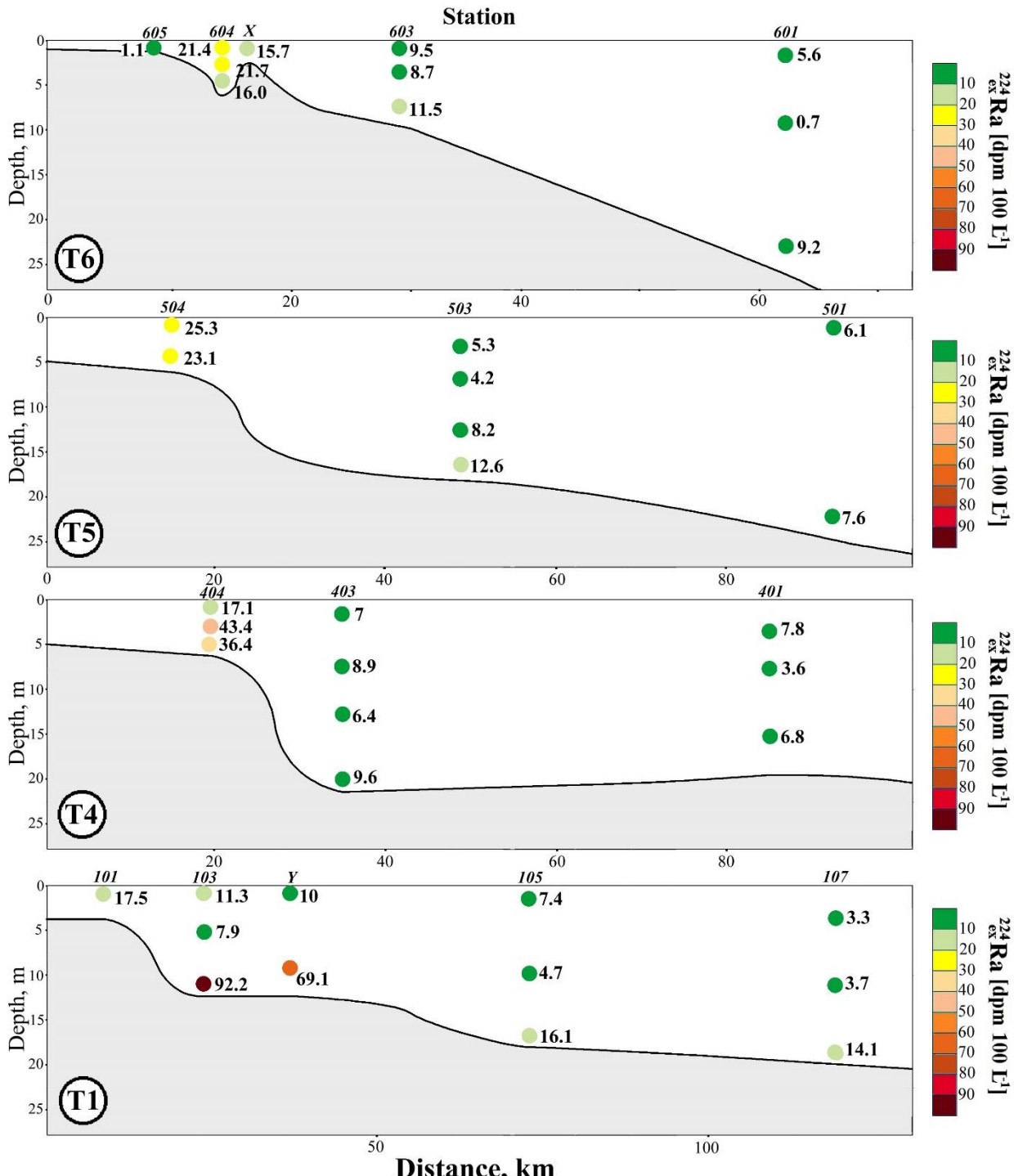

**Figure 11: Summer (September 2013) vertical distribution of ex²²⁴Ra in the Lena Delta. Sta Y (72°N, 130°E) is included in transect T1.**

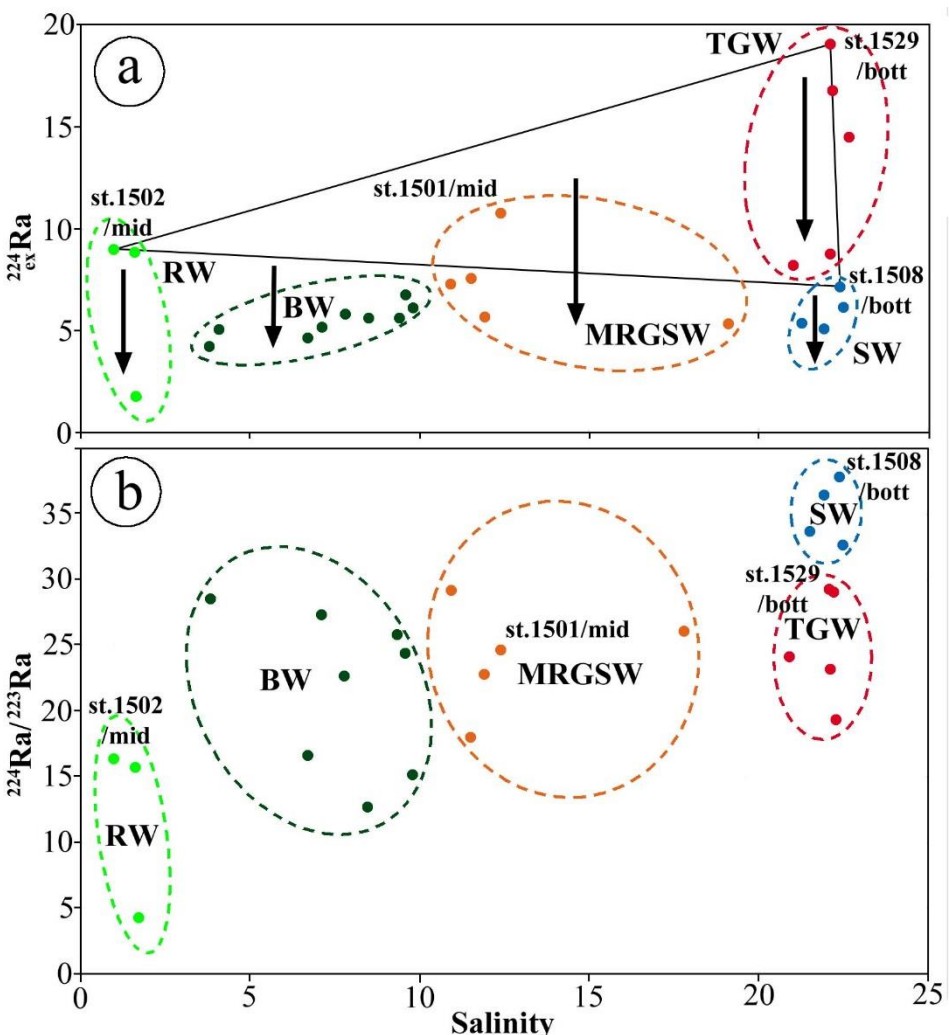

**Figure 12: Salinity vs. $^{224}$Ra (a) and $^{224}$Ra/$^{223}$Ra (b) in the winter.**

 **a) The three main water-mass endmembers are identified in this diagram: River water (RW - light green dots), seawater (SW - blue dots), and transformed ground water (TGW - red dots), and two derivatives from the main water mass because of mixing: mixed river and seawater (BW - dark green dots), and mixed RW, GW, and SW (MRGSW - yellow dots). Black arrows are theoretical decay lines.**

 **b) Similar to the mixing diagram panel (a) but using the $^{224}$Ra/$^{223}$Ra ratio. The straight line represents the theoretical mixing line derived from the major radium sources. Mid:  middle of the water column, bott: bottom.**

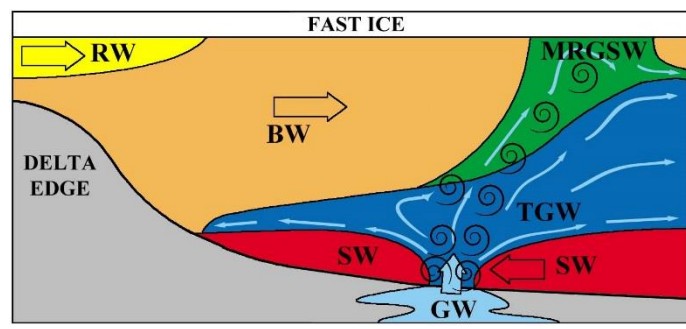

**Figure 13: The scheme of water masses mixing near the SGD location. The large black arrows show direction of endmembers movement, small light blue arrows show direction of GW flow movement, black spirals show mixing.**

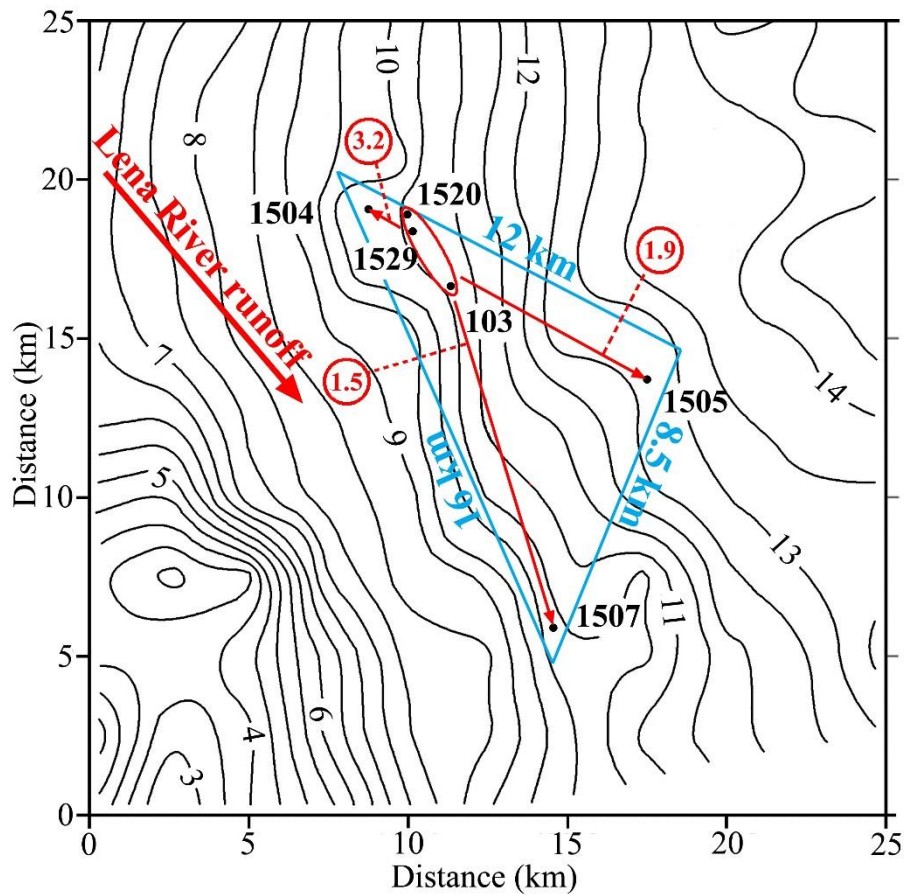

**Figure 14: Scheme of box model mass-balance mixing for SGD in the studied area. Red ellipse - the proposed SGD location. Long red arrows show the direction of TGW plume movement, numbers in red circles show the residence time (days), blue triangle indicates the base of the prism for the box model.**

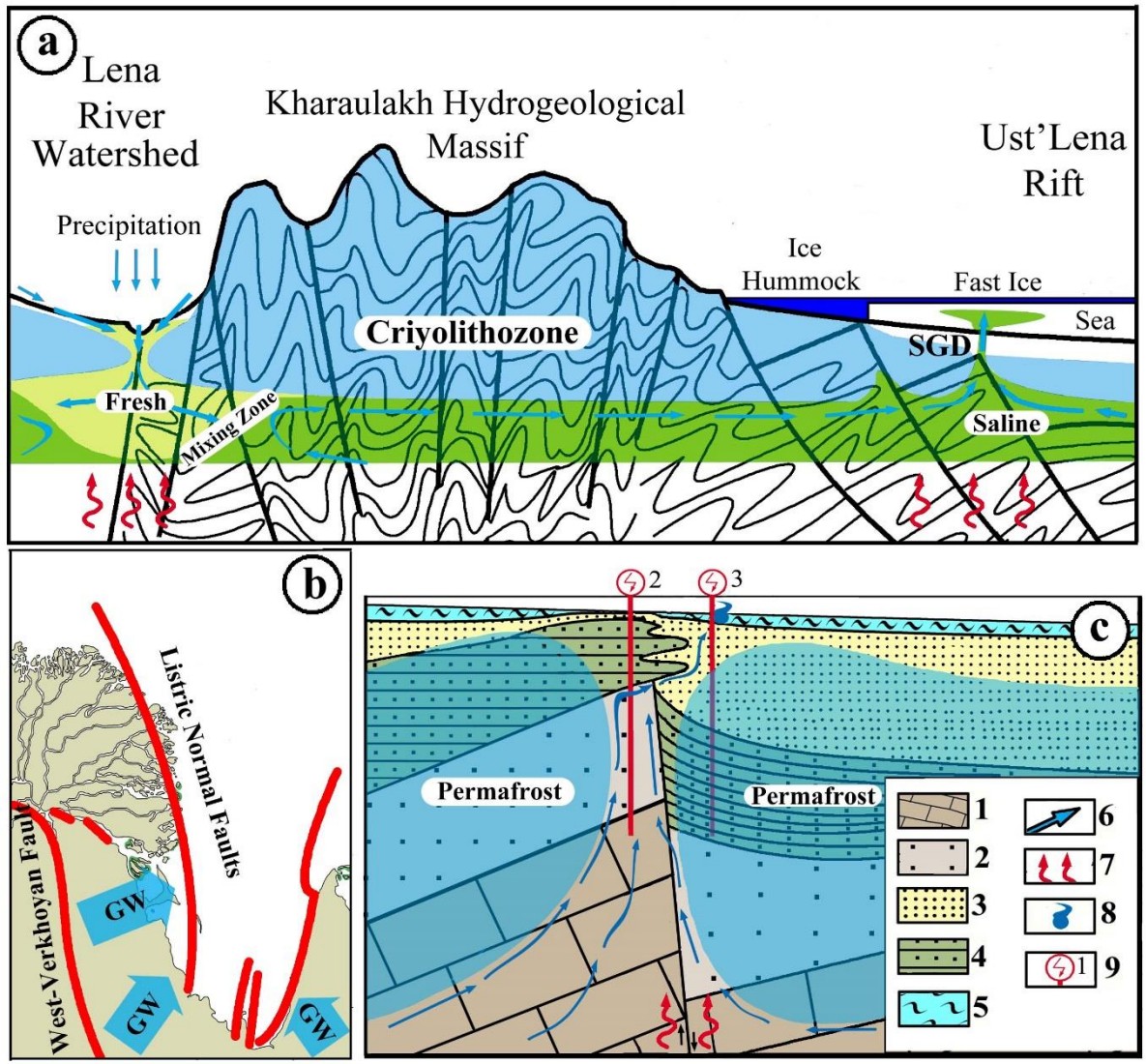

**Figure 15: The mechanistic scheme of recharge, transport, and discharge of pressurized submarine GW.**

**a) The general vertical scheme of recharge, transport, and discharge of pressurized submarine GW.**

**b) The spatial scheme of GW transport in the Kharaulakh hydrogeological massif.**

**c) The vertical scheme of GW transport and submarine discharge instead of tectonogenic talik based on TEM and drilling data. 1) Metamorphic bedrock of Pre-Upper Cretaceous complexes, 2) substratum, 3) sands, 4) aeolian Quaternary material, 5) recent bottom sediments, 6) direction of GW movement, 7) Geothermal flux, 8) SGD, 9) place and number of TEM measurements.**

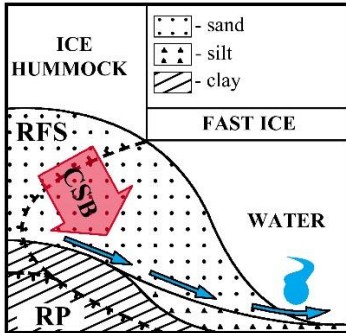

**Figure 16: Scheme of CSB and expulsion of water-soluble salts from the ice-formation zone into warmer, unfrozen parts of the sediments. RFS – recently frozen soil.  RP- relict permafrost.**

