# Peer review of "Discovery and characterization of submarine groundwater discharge in the Siberian Arctic seas: A case study in the Buor-Khaya Gulf, Laptev Sea"

_The Cryosphere, 2017_

## Referee Comment (RC1) · Anonymous Referee #1 · 13 Apr 2017

Only since about 20 years ago have ocean scientists fully appreciated the potential for submarine groundwater discharge to supply substantial quantities of nutrients and carbon to the coastal ocean. Here Charkin and co workers provide exciting new data on the potential for SGD to contribute material fluxes to the coastal Arctic Ocean, an ocean basin that is arguably undergoing the most significant changes due to climate shifts. The paper is generally well written and the topic is timely, for the reasons above. I have only one major criticism, and that is the paper is much too qualitative, given that the authors appear to have sufficient data to try and calculate SGD fluxes for this region. Perhaps the authors were rushed in their analysis of the data set in order to meet a

deadline for this special issue? In any case, the data are underutilized for reasons that are note fully explained. Other comments are listed in order of appearance:

-Abstract line 20: this sentence mentions freshwater then SGD, and we know that SGD often includes only a minor fraction of freshwater. The rest of the manuscript is good about making this distinction, but the first sentence should be reworked regardless.

-p. 3 line 17: The methods here talk about the Ra quartet, but only the short-lived Ra isotope data are presented in Table 1. One 226 and 228 value each are cited on p. 8 lines 23-24, so clearly these data exist, but it's unclear why they're not used the paper or presented in the table. Please use and publish these data!

-p. 5 methods: Where the groundwater samples (and surface water for that matter) filtered or unfiltered? If unfiltered, I am concerned about contamination of the short-lived isotopes from particulate Th isotopes (228 and 227).

-p. 7 results: the radon data are hardly used in the manuscript.

-p. 8, line 12: sediment diffusion could supply short-lived Ra isotopes to the bottom water. How is it "clear" that SGD can be the only source? Please provide a calculation to support this statement.

p. 8, section 3.4; The short lived isotopes can be highly modified by decay in addition to mixing. The linear mixing lines in Figure 11 are deceiving.

Fig. 9: axis labels are unreadable as is the legend.

Fig. 11: Salinity is the dependent variable, it should be on the x-axis.
* * *

---

## Referee Comment (RC2) · Anonymous Referee #2 · 14 Apr 2017

General comments: This paper is the first to provide direct evidence of submarine groundwater discharge in the Arctic, which is an important contribution to our understanding of the Arctic system and how it may respond to climate change. Because of this exciting new finding I recommend that this paper be published after revisions to improve the clarity of the discussion and methods.

Specific comments:

1. Introduction: The background on SGD in the Arctic is lacking, and expanding upon this will help place the importance of the current study in context. There are a few

other references that support the existence of groundwater discharge in regions of continuous permafrost based on thermal gradients (Deming et al., 1992), the mapping of springs (Kane et al., 2013), and modeling of permafrost extent taking into account freshwater inputs from SGD (Frederick and Buffett, 2015). Also, the year for Walvoord and Striegl should be 2007, not 2000.

2. p. 3 lines 24-27: Missing/incorrect references in discussion of previous studies of Ra in the Arctic: Kadko and Muench (2005) were the first to measure 224Ra in the Arctic but are not included in the list, Kadko and Aagaard (2009) did not report any short lived isotopes, and Smith et al. (2003) report 228Ra and 226Ra activities for the Beaufort Sea and central Arctic. Radium-228 activities are also reported in Trimble et al. (2004) and Cochran et al. (1995), although the main focus of these two papers is on Th and not Ra.

3. p. 5 line 21: Why were the samples not counted a third time to correct for 227Ac? If this contribution is assumed to be negligible this should be noted in the text. Clarify why total 223Ra is used instead of excess.

4. p. 7 line 38: There is no mention of how 226Ra or 228Ra are measured, but these long-lived isotopes show up later in the manuscript. The first mention of 226Ra is in the section 3.3, where it is stated that 222Rn has been corrected for ingrowth from 226Ra, but there is no explanation of how this is done. Radium-228 and 226Ra activities are also mentioned later in this section, but there is no explanation of how they are measured. If the 228Ra and 226Ra measurements were made it would be great if this data could be published, even if they long-lived isotopes are not the focus of this study!

5. p. 9 line 9: In the description of the river water endmember it is stated that the activities of 224Ra, 223Ra, and 222Rn are higher than those in seawater, but the average 224Ra in RW is less than that of SW.

6. p. 9 line 17: In the SW description it says that the 228Th/227Th ratio increases by

ingrowth. Should this say increases by decay instead of ingrowth? My understanding is that the ratio increasing because Th becomes adsorbed to the particles and then the 227Th decays faster than the 228Th while the particles are sitting in the bottom nepheloid layer.

7. p. 9: Section 3.4 could be better organized; it's a bit hard to follow the way it's written because the descriptions of the endmembers are mixed in with the interpretations of the data. It would be better if the endmember descriptions were first, and then the data were discussed in the context of the two figures (11a and 11b) separately. As is, there is really no discussion of figure 11b.

8. p. 10 line 28: Figure 12c is referenced, but I think this should be a reference to figure 12d? I recommend introducing this figure (12d) in section 3.4 instead of section 3.5.1 (make it a separate figure), because this helps in the interpretation/understanding of the endmember descriptions.

9. p.11 line 18: Was permafrost thaw considered as a source of Ra? The source of the high Ra is at the place of contact between the ice hummocks and bottom sediments, so if the ice hummocks are thawing this would be a logical place to have some runoff of the melted ice, which could be enriched in Ra.

10. It would be helpful to compare the magnitude of the discharge near Cape Muostakh to that near the Kharaulakh hydrogeological massif; this comparison might aid in the differentiation of the two discharge mechanisms.

11. Why is supplementary table 2 (which is incorrectly labeled as supplementary table 1) considered supplementary and not included in the main text? In my opinion if the wintertime data are included in the main text, the summertime data should be included as well.

Technical comments:

1. Figure 7: numbers need to be larger (can barely read contours, can't read colorbar

scales for salinity/density easily), map needs to be larger (can't read labels).

2. Figure 13: cryogenic squeezing out of brine is labeled as CSB in the caption but CSW in the figure. Recently frozen soil is labeled as RFS in the caption but RFP in the figure.

References:

Cochran, J. K., Hirschberg, D. J., Livingston, H. D., Buesseler, K. O. and Key, R. M.: Natural and anthropogenic radionuclide distributions in the Nansen Basin, Artic Ocean: Scavenging rates and circulation timescales, Deep Sea Res. Part II Top. Stud. Oceanogr., 42(6), 1495–1517, doi:10.1016/0967-0645(95)00051-8, 1995.

Deming, D., Sass, J. H., Lachenbruch, A. H. and De Rito, R. F.: Heat flow and subsurface temperature as evidence for basin-scale ground-water flow, North Slope of Alaska, Geol. Soc. Am. Bull., 104(5), 528, doi:10.1130/0016-7606(1992)104<0528:HFASTA>2.3.CO;2, 1992.

Frederick, J. M. and Buffett, B. A.: Effects of submarine groundwater discharge on the present-day extent of relict submarine permafrost and gas hydrate stability on the Beaufort Sea continental shelf, J. Geophys. Res. F Earth Surf., 120(3), 417–432, doi:10.1002/2014JF003349, 2015.

Kadko, D. and Aagaard, K.: Glimpses of Arctic Ocean shelf–basin interaction from submarine-borne radium sampling, Deep Sea Res. Part I Oceanogr. Res. Pap., 56, 32–40, doi:10.1016/j.dsr.2008.08.002, 2009.

Kadko, D. and Muench, R.: Evaluation of shelf–basin interaction in the western Arctic by use of short-lived radium isotopes: The importance of mesoscale processes, Deep Sea Res. Part II Top. Stud. Oceanogr., 52, 3227–3244, doi:10.1016/j.dsr2.2005.10.008, 2005. Kane, D. L., Yoshikawa, K. and McNamara, J. P.: Regional groundwater flow in an area mapped as continuous permafrost, NE Alaska (USA), Hydrogeol. J., 21, 41–52, doi:10.1007/s10040-012-0937-0, 2013.

Smith, J. ., Moran, S. . and Macdonald, R. .: Shelf–basin interactions in the Arctic Ocean based on 210Pb and Ra isotope tracer distributions, Deep Sea Res. Part I Oceanogr. Res. Pap., 50, 397–416, doi:10.1016/S0967-0637(02)00166-8, 2003.

Trimble, S. M., Baskaran, M. and Porcelli, D.: Scavenging of thorium isotopes in the Canada Basin of the Arctic Ocean, Earth Planet. Sci. Lett., 222(3–4), 915–932, doi:10.1016/j.epsl.2004.03.027, 2004.
* * *

---

## Referee Comment (RC3) · Anonymous Referee #3 · 20 Apr 2017

The manuscript Discovery and characterization of submarine groundwater discharge in the Siberian Arctic seas: A case study in Buor-Khaya Gulf, Laptev Sea coauthored by Charkin et al. presents the first evidences of the presence of SGD along the Eurasian Arctic margin. In my opinion the data provided in this manuscript is really interesting and will be the first step for other new studies. The paper is well written and the arguments are clear and well presented, although some parts of the text are difficult to follow. I recommend the publication of the manuscript after include some comments to the text. I have two major comments to the manuscript:

[Figure]

o Why the study is so descriptive? The text describes the distribution of short-lived Ra isotopes and Rn but does not go deeply in the use of these radionuclides as a traces to estimate SGD fluxes or transit times.

o The second comment is for me the one that I can not understand. Where is the long-lived Ra data? I guess the authors have them and maybe they want to use in another manuscript. I feel that the long-lived Ra data will help to understand SGD sources and discharge processes. For example, recently Rodellas et al (GCA, 2017) have published how the combination of short- and long-lived Ra isotopes can be used to distinguish sources of submarine groundwater discharge: fresh groundwater vs seawater recirculation through sediments that is the case of part of the study. How you can discard that the short-lived Ra isotopes are not coming from recirculation or sediment resuspension?

Some other technical corrections are:

o Do you have any sample for Ra-isotopes and Rn that can be compared between winter and summer? o Line 35 pag 5. Do you need the two dots after Radiometer?

o Review the text because sometimes you write "kilometers" or "meters" and others you write "km" and "m".

o I suggest rewriting the sentence of the Line 20 Pag 6 as: These TEM results agree well with data provided by Imaev et al (2004) for this region.

o Line 22 Page 6. Remove a space between 162, and 5 meters.

o 3.2 Features of the thermohaline water structure and SGD fate. In this chapter authors explained the features of the thermohaline water structure but there is not any comparison or relation with SGD. Is this a correct tittle?

o In the chapter 3.3 the units of Ra-223, Ra-224 and Rn are wrong. It is "dpmÂů100L-1" or "dpm/100L". Check the document and Figures.
o Line 25 Page 8. Add a space between the numbers and "m" as you do for "km" or other units.

o Line 10 to 13 Pag 8. Maybe the 228/226 AR provides information about the SGD fluxes.

o Line 10 page 9. Maybe this is a possible explanation, but what about the increase due to resuspension processes? Is it a possibility? Maybe the long-lived Ra isotopes can help you to understand the origin of this high 224/223 AR.

o Line 33 page 9. Correct the 100L-1.

o Line 20 page 10. Can you explain why this water is saltier?

o Line 38 page 10. Here you notice: The fact that the summertime and wintertime SGD springs were found on a line parallel to the fault once again points to the connection between the tectonogenic talik and the SGD. Here you talk about summertime and wintertime SGD springs. Why the seasonality is not shown in the text?

o Table 1. Add the uncertainties to the 224/223 AR and review the significant figures of the Ra-223 and Ra-224.

o Figure 11. What is the meaning of the equation of the upper plot?

o Figure 13. I guess there is a mistake in the Figure referred to CSW in the plot and RFS in the caption.

---

## Author Comment (AC1) · 16 Jun 2017

We thank all three reviewers for their large effort and for providing valuable and very constructive comments, which have been useful in our revisions of the manuscript. Naturally, we are encouraged that all reviewers support our study of submarine ground-water discharge (SGD) in the Bhuor-Khaya Bay, SE Laptev Sea, and the conclusion that it provides a previously largely unexplored vector for transport from land to the East Siberian Arctic shelf, yet complicated by geocryological conditions such as permafrost. Below, each review comment is listed first, followed by our response and a description

of resulting edit. Author comments are marked below as AC.

General comments by anonymous referee 1 (Reviewer 1).

RC: Only since about 20 years ago have ocean scientists fully appreciated the potential for submarine groundwater discharge to supply substantial quantities of nutrients and carbon to the coastal ocean. Here Charkin and co workers provide exciting new data on the potential for SGD to contribute material fluxes to the coastal Arctic Ocean, an ocean basin that is arguably undergoing the most significant changes due to climate shifts. The paper is generally well written and the topic is timely, for the reasons above. I have only one major criticism, and that is the paper is much too qualitative, given that the authors appear to have sufficient data to try and calculate SGD fluxes for this region. Perhaps the authors were rushed in their analysis of the data set in order to meet a deadline for this special issue? In any case, the data are underutilized for reasons that are note fully explained.

AC: Thank you for your appreciation of our work. We were originally hesitant to make too much quantitative calculations because of the limited database, requiring many assumptions. Following the reviewer encouragement, we have now added an estimate of SGD discharge (1.7 x 10 6 m3 d-1) and transit times (3.2 – 1.5 days) around the SGD place, while stating clearly all assumptions. Discharge of the subpermafrost groundwater from Kharaulakh hydrogeological massif through the talik area were calculated on excess 224Ra activities using a Ra mass balance model (Moore 1996; Burnett and Dulaiova 2003; Null et. al., 2012, 2014). In order to calculate the growth of the water mass "radium ages," we used the equation proposed by Moore (2000). As a result, there will be two new chapters in the methods section and one in the discussion section. This results will be shown in our revised text.

Specific comments by Reviewer 1

RC: Abstract line 20: this sentence mentions freshwater then SGD, and we know that SGD often includes only a minor fraction of freshwater. The rest of the manuscript is

good about making this distinction, but the first sentence should be reworked regardless.

AC: Thanks, we reworked this sentence.

RC: p. 3 line 17: The methods here talk about the Ra quartet, but only the short-lived Ra isotope data are presented in Table 1. One 226 and 228 value each are cited on p. 8 lines 23-24, so clearly these data exist, but it's unclear why they're not used the paper or presented in the table. Please use and publish these data!

AC: We would have liked to publish wintertime data for long-lived Ra isotopes, but this is beyond the scope of the present study. Once/if such gamma counting results of long-lived radium isotopes for wintertime will be delivered from our collaborators at the Radium Institute (located in Sankt-Petersburg), it will be included in a future study. So, we have to keep Table 1 with no changes. However, in the final version of the manuscript we will include data on long-lived isotopes in the summertime: Regarding methods for Ra isotopes, this will be included in the revised ms. Briefly, in the shore-based/home laboratory, Ra was leached from the fibre with hot 6N HCl, coprecipitated as $BaSO_4$ and counted with gamma spectroscopy for 226Ra and 228Ra (Moore, W.S., 1984. Radium isotope measurements using germanium detectors. Nuclear Instruments and Methods in Physics Research 223, 407-411). So, we modify Table 1/Suppl materials with the summer data for long-lived Ra isotopes; and move this table to the main text.

RC: p. 5 methods: Where the groundwater samples (and surface water for that matter) filtered or unfiltered? If unfiltered, I am concerned about contamination of the shortlived isotopes from particulate Th isotopes (228 and 227).

AC: All samples were passed over Hytrex cartridges with $1\mu$m nominal pore size. We will add this information to the methods p.5, line 14.

RC: p. 7 results: the radon data are hardly used in the manuscript.

[Figure]

AC: As can be seen from the table 1, we have much less radon data compared to the radium isotopes. This is a reason why the radon data is used less in the manuscript.

RC: p. 8, line 12: sediment diffusion could supply short-lived Ra isotopes to the bottom water. How is it "clear" that SGD can be the only source? Please provide a calculation to support this statement.

AC: We do not have data on the production rate of 224Ra (and 223Ra) within the surface sediment, so we see no way to make a reliable estimate of the diffusive flux of 224Ra out of the sediment. The approach developed by Nozaki and applied by Moore confirms the low ratio of long-lived to short-lived nuclides in the diffusive flux, which we used as argument to separate the role of diffusion and SGD in the summertime cruise. These results will be shown in our revised text.

RC: p. 8, section 3.4; The short lived isotopes can be highly modified by decay in addition to mixing. The linear mixing lines in Figure 11 are deceiving.

AC: Yes, we agree. We will remove the linear mixing lines and add the expected decay lines (for better perception) to the plot.

RC: Fig. 9: axis labels are unreadable as is the legend.

AC: This figure will be modifed accordingly.

RC: Fig. 11: Salinity is the dependent variable; it should be on the x-axis.

AC: You are right. This will be fixed.

Thank you for your valuable comments which help to improve our manuscript.

———————————————

---

## Author Comment (AC2) · 16 Jun 2017

We thank all three reviewers for their large effort and for providing valuable and very constructive comments, which have been useful in our revisions of the manuscript. Naturally, we are encouraged that all reviewers support our study of submarine ground-water discharge (SGD) in the Bhuor-Khaya Bay, SE Laptev Sea, and the conclusion that it provides a previously largely unexplored vector for transport from land to the East Siberian Arctic shelf, yet complicated by geocryological conditions such as permafrost. Below, each review comment is listed first, followed by our response and a description

of resulting edit. Author comments are marked below as AC.

General comments by anonymous referee 2 (Reviewer 2).

RC: General comments: This paper is the first to provide direct evidence of submarine groundwater discharge in the Arctic, which is an important contribution to our understanding of the Arctic system and how it may respond to climate change. Because of this exciting new finding I recommend that this paper be published after revisions to improve the clarity of the discussion and methods.

AC: Thanks for the appreciation of our manuscript.

Specific comments by Reviewer 2

RC: Introduction: The background on SGD in the Arctic is lacking, and expanding upon this will help place the importance of the current study in context. There are a few C1 TCD Interactive comment Printer-friendly version Discussion paper other references that support the existence of groundwater discharge in regions of continuous permafrost based on thermal gradients (Deming et al., 1992), the mapping of springs (Kane et al., 2013), and modeling of permafrost extent taking into account freshwater inputs from SGD (Frederick and Buffett, 2015). Also, the year for Walvoord and Striegl should be 2007, not 2000.

AC: Thanks for the constructive remark. We will edit the introduction accordingly - using the proposed literature and correct the year for Walvoord and Striegl.

RC: p. 3 lines 24-27: Missing/incorrect references in discussion of previous studies of Ra in the Arctic: Kadko and Muench (2005) were the first to measure 224Ra in the Arctic but are not included in the list, Kadko and Aagaard (2009) did not report any short lived isotopes, and Smith et al. (2003) report 228Ra and 226Ra activities for the Beaufort Sea and central Arctic. Radium-228 activities are also reported in Trimble et al. (2004) and Cochran et al. (1995), although the main focus of these two papers is on Th and not Ra.

AC: Thank you. We will edit accordingly.

RC: p. 5 line 21: Why were the samples not counted a third time to correct for 227Ac? If this contribution is assumed to be negligible this should be noted in the text. Clarify why total 223Ra is used instead of excess.

AC: Yes, the samples were counted a third time to correct for 227 Ac. The Ac activity was in the error range. Thus, the Ac contribution is assumed to be negligible. We will note this in the revised text.

RC: p. 7 line 38: There is no mention of how 226Ra or 228Ra are measured, but these long-lived isotopes show up later in the manuscript. The first mention of 226Ra is in the section 3.3, where it is stated that 222Rn has been corrected for ingrowth from 226Ra, but there is no explanation of how this is done. Radium-228 and 226Ra activities are also mentioned later in this section, but there is no explanation of how they are measured. If the 228Ra and 226Ra measurements were made it would be great if this data could be published, even if they long-lived isotopes are not the focus of this study!

AC: We still do not have data on long-lived isotopes for wintertime (see our detailed response to a similar question by Rev. 1). However, in the final version of the manuscript we will include data on long-lived isotopes in the summertime: Regarding methods for Ra isotopes, this will be included in the revised ms. Briefly, in the shorebased/home laboratory, Ra was leached from the fibre with hot 6N HCl, coprecipitated as BaSO4 and counted with gamma spectroscopy for 226Ra and 228Ra (Moore, W.S., 1984. Radium isotope measurements using germanium detectors. Nuclear Instruments and Methods in Physics Research 223, 407-411).

RC: p. 9 line 9: In the description of the river water endmember it is stated that the activities of 224Ra, 223Ra, and 222Rn are higher than those in seawater, but the average 224Ra in RW is less than that of SW.

[Figure]

AC: In this paper we consider three freshest samples as the riverine (RW), but to choose the "best" end-member we use 222Rn, 223 Ra, 224Ra obtained at the station 1502 which is characterized by the lowest salinity (0.98psu).

RC: p. 9 line 17: In the SW description it says that the 228Th/227Th ratio increases by ingrowth. Should this say increases by decay instead of ingrowth? My understanding is that the ratio increasing because Th becomes adsorbed to the particles and then the 227Th decays faster than the 228Th while the particles are sitting in the bottom nepheloid layer.

AC: Yes, you are right. In principle, we had this in mind, but a little confused, because we are not native speakers of English. This will be corrected.

RC: p. 9: Section 3.4 could be better organized; it's a bit hard to follow the way it's written because the descriptions of the endmembers are mixed in with the interpretations of the data. It would be better if the endmember descriptions were first, and then the data were discussed in the context of the two figures (11a and 11b) separately. As is, there is really no discussion of figure 11b.

AC: We agree; we will re-organize this section and add more discussion about figure 11b. Thank you.

RC: p. 10 line 28: Figure 12c is referenced, but I think this should be a reference to figure 12d? I recommend introducing this figure (12d) in section 3.4 instead of section 3.5.1 (make it a separate figure), because this helps in the interpretation/understanding of the endmember descriptions.

AC: Yes, correct, it should be 12d. It is a typo, which will be corrrected. Yes, you are right, it is better to make this figure separately and include it in section 3.4. We will revise the ms to this effect.

RC: 9. p.11 line 18: Was permafrost thaw considered as a source of Ra? The source of the high Ra is at the place of contact between the ice hummocks and bottom sediments,

so if the ice hummocks are thawing this would be a logical place to have some runoff of the melted ice, which could be enriched in Ra.

AC: Runoff of the melted ice is unlikely, because during our wintertime studies, the air temperature did not rise above -10 degrees Celsius during the day, and at night, it dropped to -30. The temperature of the water was negative everywhere. Moreover, the high salinity of the waters along the ice hummocks periphery also indicates cryogenic squeezing out of brine and water-soluble salts as plausible mechanism of the radium enrichment. We will seek to clarify this information further in the revised text

RC: It would be helpful to compare the magnitude of the discharge near Cape Muostakh to that near the Kharaulakh hydrogeological massif; this comparison might aid in the differentiation of the two discharge mechanisms.

AC: In the final version of this paper, we plan to show calculations of SGD discharge from the Kharaulakh hydrogeological massif and transit times using 224Ra and 223Ra. However, we have not enough statistics (only one station) to calculate the magnitude of discharge near Cape Muostakh, so this is a task for our future research.

RC: Why is supplementary table 2 (which is incorrectly labeled as supplementary table 1) considered supplementary and not included in the main text? In my opinion if the wintertime data are included in the main text, the summertime data should be included as well.

AC: We will move this table from suppl materials into the main text and add there the data on long-lived isotopes. Thank you.

Technical comments:

RC: Figure 7: numbers need to be larger (can barely read contours, can't read colorbar scales for salinity/density easily), map needs to be larger (can't read labels).

AC: We will rework figure 7 to increase visibility and clarity.

RC: Figure 13: cryogenic squeezing out of brine is labeled as CSB in the caption but CSW in the figure. Recently frozen soil is labeled as RFS in the caption but RFP in the figure.

AC: Sorry. It is a typo. We will edit the figure.

Thank you for your valuable comments which help to improve our manuscript.

---

## Author Comment (AC3) · 16 Jun 2017

We thank all three reviewers for their large effort and for providing valuable and very constructive comments, which have been useful in our revisions of the manuscript. Naturally, we are encouraged that all reviewers support our study of submarine groundwater discharge (SGD) in the Bhuor-Khaya Bay, SE Laptev Sea, and the conclusion that it provides a previously largely unexplored vector for transport from land to the East Siberian Arctic shelf, yet complicated by geocryological conditions such as permafrost. Below, each review comment is listed first, followed by our response and a description

of resulting edit. Author comments are marked below as AC.

General and specific comments by anonymous referee 3 (Reviewer 3).

RC: The manuscript Discovery and characterization of submarine groundwater discharge in the Siberian Arctic seas: A case study in Buor-Khaya Gulf, Laptev Sea coauthored by Charkin et al. presents the first evidences of the presence of SGD along the Eurasian Arctic margin. In my opinion the data provided in this manuscript is really interesting and will be the first step for other new studies. The paper is well written and the arguments are clear and well presented, although some parts of the text are difficult to follow. I recommend the publication of the manuscript after include some comments to the text. I have two major comments to the manuscript:

Why the study is so descriptive? The text describes the distribution of short-lived Ra isotopes and Rn but does not go deeply in the use of these radionuclides as a traces to estimate SGD fluxes or transit times.

AC: Thank you for the overall support and for this specific comment. We have now in the revised ms, incorporated a quantitative estimation of SGD discharge and transit times using short-lived Ra isotopes. Following the reviewers encouragement, we have now added an estimate of SGD discharge (1.7 x 10 6 m3 d-1) and transit times (3.2 – 1.5 days) around the SGD place, while stating clearly all assumptions. Discharge of the subpermafrost groundwater from Kharaulakh hydrogeological massif through the talik area were calculated on excess 224Ra activities using a Ra mass balance model (Moore 1996; Burnett and Dulaiova 2003; Null et. al., 2012, 2014). In order to calculate the growth of the water mass "radium ages," we used the equation proposed by Moore (2000). As a result, there will be two new chapters in the methods section and one in the discussion section. This results will be shown in our revised text.

RC: The second comment is for me the one that I can not understand. Where is the long-lived Ra data? I guess the authors have them and maybe they want to use in another manuscript. I feel that the long-lived Ra data will help to understand SGD

sources and discharge processes. For example, recently Rodellas et al (GCA, 2017) have published how the combination of short- and long-lived Ra isotopes can be used to distinguish sources of submarine groundwater discharge: fresh groundwater vs seawater recirculation through sediments that is the case of part of the study.

AC: We agree with this comment, but we still have not obtained the results of gamma spectrometry for the long-lived Ra isotopes for the wintertime (see our detailed response to a similar question by Rev. 1). However, in the final version of the manuscript we will include data on long-lived isotopes in the summertime: Regarding methods for Ra isotopes, this will be included in the revised ms. Briefly, in the shorebased/home laboratory, Ra was leached from the fibre with hot 6N HCl, coprecipitated as BaSO4 and counted with gamma spectroscopy for 226Ra and 228Ra (Moore, W.S., 1984. Radium isotope measurements using germanium detectors. Nuclear Instruments and Methods in Physics Research 223, 407-411).

RC: How you can discard that the short-lived Ra isotopes are not coming from recirculation or sediment resuspension?

AC: This is a very important comment and we have now included (in section 3.3) the long lived isotope data of the summertime cruise and an additional figure of radium isotopes versus salinity as supporting argument that the high activities of short lived Ra isotopes near station 1529 are due to SGD and not to resuspension (see the figure 1). Besides, according to our multi-year winter data, the resuspension effect is negligible during wintertime (see the details elsewhere: Charkin et al., BG, 2011)

Some other technical corrections are:

RC: Do you have any sample for Ra-isotopes and Rn that can be compared between winter and summer? o Line 35 pag 5. Do you need the two dots after Radiometer?

AC: Unfortunately, we do not have such data.

RC: Review the text because sometimes you write "kilometers" or "meters" and others

you write "km" and "m".

AC: Yes, we apologize for not being consistent in use of these jnits. We will check the text carefully and streamline this

RC: I suggest rewriting the sentence of the Line 20 Pag 6 as: These TEM results agree well with data provided by Imaev et al (2004) for this region.

AC: Ok, will be done.

RC: Line 22 Page 6. Remove a space between 162, and 5 meters.

AC: Ok, will be done.

RC: 3.2 Features of the thermohaline water structure and SGD fate. In this chapter authors explained the features of the thermohaline water structure but there is not any comparison or relation with SGD. Is this a correct tittle?

AC: Yes, we agree: that is not correct. It will be changed to Features of the thermohaline water structure.

RC: In the chapter 3.3 the units of Ra-223, Ra-224 and Rn are wrong. It is "dpm ̎u100L-1" or "dpm/100L". Check the document and Figures.

AC: We are sorry, this is a typo. It should of course be dpm/100L or dpm 100L-1. It will be edited to dpm 100L-1.

RC: Line 25 Page 8. Add a space between the numbers and "m" as you do for "km" or other units.

AC: Ok, will be done.

RC: Line 10 to 13 Pag 8. Maybe the 228/226 AR provides information about the SGD fluxes.

AC: Yes, you are right. We included to the manuscript the summer data on long-lived isotopes which supporting argument that the high activities of short-lived Ra isotopes

near station 1529 are due to SGD.

RC: Line 10 page 9. Maybe this is a possible explanation, but what about the increase due to resuspension processes? Is it a possibility? Maybe the long-lived Ra isotopes can help you to understand the origin of this high 224/223 AR.

AC: Yes, you are right, this could be an explanation for summer time, but in under ice conditions at a relatively large depth and with a low winter Lena River discharge, resuspension is very unlikely. This is also indicated by the very low concentration of suspended particulate matter in the water for this time (Charkin et al. BG, 2011). Moreover, we included to the manuscript the summer data on long-lived isotopes which supporting argument that the high activities of short-lived Ra isotopes near station 1529 are due to SGD and not to resuspension (see the figure 1).

RC: Line 33 page 9. Correct the 100L-1.

AC: We will correct.

RC: Line 20 page 10. Can you explain why this water is saltier?

AC: This water is saltier because this water was formed as the result of multi-year freezing of rocks and the simultaneous concentration of salts (Pinneker, 1983; Romanovskii, 1983), resulting in elevated levels of total dissolved solids in the cryogenic groundwater.

RC: Line 38 page 10. Here you notice: The fact that the summertime and wintertime SGD springs were found on a line parallel to the fault once again points to the connection between the tectonogenic talik and the SGD. Here you talk about summertime and wintertime SGD springs. Why the seasonality is not shown in the text?

AC: The seasonality is not shown in the text, because the summer data is very limited for SGD place.

RC: Table 1. Add the uncertainties to the 224/223 AR and review the significant figures

of the Ra-223 and Ra-224.

AC: Ok, will be done.

RC: What is the meaning of the equation of the upper plot?

AC: We will redo this piece of the text by removing both the linear mixing lines (because short-lived isotopes can be highly modified by decay in addition to mixing) and the equation.

RC: I guess there is a mistake in the Figure referred to CSW in the plot and RFS in the caption.

AC: Yes, you are right. It is a typo. We will correct. Thank you.

Thank you for your valuable comments which help to improve our manuscript.

[Figure]

[Figure]

[Figure]

**Fig. 1.**

---

## Author Response (AR1)

We thank all three reviewers for their large effort and for providing valuable and very constructive comments, which have been useful in our revisions of the manuscript. Naturally, we are encouraged that all reviewers support our study of submarine groundwater discharge (SGD) in the Bhuor-Khaya Bay, SE Laptev Sea, and the conclusion that it provides a previously largely unexplored vector for transport from land to the East Siberian Arctic shelf, yet complicated by geocryological conditions such as permafrost. Below, each review comment is listed first, followed by our response and a description of resulting edit. Author comments are marked below as AC. Pages and lines of changes in the final version are shown in parentheses.

**General comments by anonymous referee 1 (Reviewer 1).**

*RC: Only since about 20 years ago have ocean scientists fully appreciated the potential for submarine groundwater discharge to supply substantial quantities of nutrients and carbon to the coastal ocean. Here Charkin and co workers provide exciting new data on the potential for SGD to contribute material fluxes to the coastal Arctic Ocean, an ocean basin that is arguably undergoing the most significant changes due to climate shifts. The paper is generally well written and the topic is timely, for the reasons above. I have only one major criticism, and that is the paper is much too qualitative, given that the authors appear to have sufficient data to try and calculate SGD fluxes for this region. Perhaps the authors were rushed in their analysis of the data set in order to meet a deadline for this special issue? In any case, the data are underutilized for reasons that are note fully explained.*

AC: Thank you for your appreciation of our work. We were originally hesitant to make too much quantitative calculations because of the limited database, requiring many assumptions. Following the reviewer encouragement, we have now added an estimate of SGD discharge 1.7 x 10$^6$ m3 d-1**(p.1, line 32; p.12, line 37; p.15, line 1)** and transit times 3.2 – 1.5 days **(p.12, line 23-31)** around the SGD place, while stating clearly all assumptions. Discharge of the subpermafrost groundwater from Kharaulakh hydrogeological massif through the talik area were calculated on excess 224Ra activities using a Ra mass balance model (Moore 1996; Burnett and Dulaiova 2003; Null et. al., 2012, 2014). In order to calculate the growth of the water mass "radium ages," we used the equation proposed by Moore (2000). As a result, there will be two new chapters **(chapter 2.5.2, p.6; chapter 2.2.3, p.7)** in the methods section and one in the discussion section **(chapter 3.5, p.11)**.

**Specific comments by Reviewer 1**

*RC: Abstract line 20: this sentence mentions freshwater then SGD, and we know that SGD often includes only a minor fraction of freshwater. The rest of the manuscript is good about making this distinction, but the first sentence should be reworked regardless.*

AC:  Thanks, we have reworked this sentence **(p.1, line 20)**.

*RC: p. 3 line 17: The methods here talk about the Ra quartet, but only the short-lived Ra isotope data are presented in Table 1. One 226 and 228 value each are cited on p. 8 lines 23-24, so clearly these data exist, but it's unclear why they're not used the paper or presented in the table. Please use and publish these data!*

AC: We would have liked to publish wintertime data for long-lived Ra isotopes, but this is beyond the scope of the present study. Once/if such gamma counting results of long-lived radium isotopes for wintertime will be delivered from our collaborators at the Radium Institute (located in Sankt-Petersburg), it will be included in a future study. So, we have to keep Table 1 with no changes.

However, in the final version of the manuscript we will include data on long-lived isotopes in the summertime **(p.9, line 26-42; p.10, line 1-4; fig.10; table 2)**:

Regarding methods for Ra isotopes, this will be included in the revised ms. Briefly, in the shorebased/home laboratory, Ra was leached from the fibre with hot 6N HCl, coprecipitated as BaSO4 and counted with gamma spectroscopy for 226Ra and 228Ra (Moore, W.S., 1984. Radium isotope measurements using germanium detectors **(p.6, line 19)**. Nuclear Instruments and Methods in Physics Research 223, 407-411). So, we modify Table 1/Suppl materials with the summer data for long-lived Ra isotopes; and move this table to the main text.

*RC: p. 5 methods: Where the groundwater samples (and surface water for that matter) filtered or unfiltered? If unfiltered, I am concerned about contamination of the shortlived isotopes from particulate Th isotopes (228 and 227).*

AC: All samples were passed over Hytrex cartridges with 1μm nominal pore size. We have added this information to the methods **(p.6, line 1)**.

*RC: p. 7 results: the radon data are hardly used in the manuscript.*

AC: As can be seen from the table 1, we have much less radon data compared to the radium isotopes. This is a reason why the radon data is used less in the manuscript. Finally, we decided to remove the radon data since we do not use them.

*RC: p. 8, line 12: sediment diffusion could supply short-lived Ra isotopes to the bottom water. How is it "clear" that SGD can be the only source? Please provide a calculation to support this statement.*

AC: We do not have data on the production rate of 224Ra (and 223Ra) within the surface sediment, so we see no way to make a reliable estimate of the diffusive flux of 224Ra out of the sediment. The approach developed by Nozaki and applied by Moore confirms the low ratio of long-lived to short-lived nuclides in the diffusive flux, which we used as argument to separate the role of diffusion and SGD in the summertime cruise **(p.9, line 26-42; p.10, line 1-4; fig.10; table 2)**.

*RC: p. 8, section 3.4; The short lived isotopes can be highly modified by decay in addition to mixing. The linear mixing lines in Figure 11 are deceiving.*

AC: Yes, we agree. We have removed the linear mixing lines and added the expected decay lines (for better perception) to the plot **(fig.12)**.

*RC: Fig. 9: axis labels are unreadable as is the legend.*

AC: This figure modified accordingly **(fig.9)**.

*RC: Fig. 11: Salinity is the dependent variable; it should be on the x-axis.*

AC: You are right. We have changed **(fig.12)**.

Thank you for your valuable comments which help to improve our manuscript.

**General comments by anonymous referee 2 (Reviewer 2).**

*RC: General comments: This paper is the first to provide direct evidence of submarine groundwater discharge in the Arctic, which is an important contribution to our understanding of the Arctic system and how it may respond to climate change. Because of this exciting new finding I recommend that this paper be published after revisions to improve the clarity of the discussion and methods.*

AC: Thanks for the appreciation of our manuscript.

**Specific comments by Reviewer 2**

*RC: Introduction: The background on SGD in the Arctic is lacking, and expanding upon this will help place the importance of the current study in context. There are a few C1 TCD Interactive comment Printer-friendly version Discussion paper other references that support the existence of groundwater discharge in regions of continuous permafrost based on thermal gradients (Deming et al., 1992), the mapping of springs (Kane et al., 2013), and modeling of permafrost extent taking into account freshwater inputs from SGD (Frederick and Buffett, 2015). Also, the year for Walvoord and Striegl should be 2007, not 2000.*

AC: Thanks for the constructive remark. We have edited the introduction accordingly - using the proposed literature **(p.3, line 9; p.3, line 34-42)** and correct the year for Walvoord and Striegl **(p.2, line 34)**.

*RC: p. 3 lines 24-27: Missing/incorrect references in discussion of previous studies of Ra in the Arctic: Kadko and Muench (2005) were the first to measure 224Ra in the Arctic but are not included in the list, Kadko and Aagaard (2009) did not report any short lived isotopes, and Smith et al. (2003) report 228Ra and 226Ra activities for the Beaufort Sea and central Arctic. Radium-228 activities are also reported in Trimble et al. (2004) and Cochran et al. (1995), although the main focus of these two papers is on Th and not Ra.*

AC: Thank you. We have edited accordingly **(p.4, line 8-11)**.

*RC: p. 5 line 21: Why were the samples not counted a third time to correct for 227Ac? If this contribution is assumed to be negligible this should be noted in the text. Clarify why total 223Ra is used instead of excess.*

AC: Yes, the samples were counted a third time to correct for 227 Ac. The Ac activity was below detection limit. Thus, the Ac contribution is assumed to be negligible **(p.6, line 10)**.

*RC: p. 7 line 38: There is no mention of how 226Ra or 228Ra are measured, but these long-lived isotopes show up later in the manuscript. The first mention of 226Ra is in the section 3.3, where it is stated that 222Rn has been corrected for ingrowth from 226Ra, but there is no explanation of how this is done. Radium-228 and 226Ra activities are also mentioned later in this section, but there is no explanation of how they are measured. If the 228Ra and 226Ra measurements were made it would be great if this data could be published, even if they long-lived isotopes are not the focus of this study!*

AC: We still do not have data on long-lived isotopes for wintertime (see our detailed response to a similar question by Rev. 1). However, in the final version of the manuscript we will include data on long-lived isotopes in the summertime **(p.9, line 26-42; p.10, line 1-4; fig.10; table 2)**:

Regarding methods for Ra isotopes, this will be included in the revised ms. Briefly, in the shorebased/home laboratory, Ra was leached from the fibre with hot 6N HCl, coprecipitated as BaSO4 and counted with gamma spectroscopy for 226Ra and 228Ra (Moore, W.S., 1984. Radium isotope measurements using germanium detectors **(p.6, line 19)**. Nuclear Instruments and Methods in Physics Research 223, 407-411).

*RC: p. 9 line 9: In the description of the river water endmember it is stated that the activities of 224Ra, 223Ra, and 222Rn are higher than those in seawater, but the average 224Ra in RW is less than that of SW.*

AC: In this paper we consider three freshest samples as the riverine (RW), but to choose the "best" end-member we use 223 Ra and 224Ra obtained at the station 1502 which is characterized by the lowest salinity (0.98psu)**(table 1).**

*RC: p. 9 line 17: In the SW description it says that the 228Th/227Th ratio increases by ingrowth. Should this say increases by decay instead of ingrowth? My understanding is that the ratio increasing because Th becomes adsorbed to the particles and then the 227Th decays faster than the 228Th while the particles are sitting in the bottom nepheloid layer.*

AC: Yes, you are right. In principle, we had this in mind, but a little confused, because we are not native speakers of English. We have corrected **(p.11, line 16)**.

*RC: p. 9: Section 3.4 could be better organized; it's a bit hard to follow the way it's written because the descriptions of the endmembers are mixed in with the interpretations of the data. It would be better if the endmember descriptions were first, and then the data were discussed in the context of the two figures (11a and 11b) separately. As is, there is really no discussion of figure 11b.*

AC: We agree; we have re-organized this section and added more discussion about figure 11b. Thank you. See the new version of the section on the **p.10-11**.

*RC: p. 10 line 28: Figure 12c is referenced, but I think this should be a reference to figure 12d? I recommend introducing this figure (12d) in section 3.4 instead of section 3.5.1 (make it a separate figure), because this helps in the interpretation/understanding of the endmember descriptions.*

AC: Yes, correct, it should be 12d. It is a typo, which have corrected. Yes, you are right, it is better to make this figure separately and include it in section 3.4. **(New fig.13)**.

*RC: 9. p.11 line 18: Was permafrost thaw considered as a source of Ra? The source of the high Ra is at the place of contact between the ice hummocks and bottom sediments, so if the ice hummocks are thawing this would be a logical place to have some runoff of the melted ice, which could be enriched in Ra.*

AC: Runoff of the melted ice is unlikely, because during our wintertime studies, the air temperature did not rise above -10 degrees Celsius during the day, and at night, it dropped to -30. The temperature of the water was negative everywhere. Moreover, the high salinity of the waters along the ice hummocks periphery also indicates cryogenic squeezing out of brine and water-soluble salts as plausible mechanism of the radium enrichment. We clarified this information in the revised text **(p.14, line 12)**.

*RC: It would be helpful to compare the magnitude of the discharge near Cape Muostakh to that near the Kharaulakh hydrogeological massif; this comparison might aid in the differentiation of the two discharge mechanisms.*

AC: In the final version of this paper, we plan to show calculations of SGD discharge from the Kharaulakh hydrogeological massif and transit times using 224Ra and 223Ra. However, we have not enough statistics (only one station) to calculate the magnitude of discharge near Cape Muostakh, so this is a task for our future research.

*RC: Why is supplementary table 2 (which is incorrectly labeled as supplementary table 1) considered supplementary and not included in the main text? In my opinion if the wintertime data are included in the main text, the summertime data should be included as well.*

AC: We have moved this table from suppl materials into the main text and added there the data on long-lived isotopes **(New table 2)**. Thank you.

**Technical comments:**

*RC: Figure 7: numbers need to be larger (can barely read contours, can't read colorbar scales for salinity/density easily), map needs to be larger (can't read labels).*

AC: We have reworked figure 7 to increase visibility and clarity **(fig.7)**.

*RC: Figure 13: cryogenic squeezing out of brine is labeled as CSB in the caption but CSW in the figure. Recently frozen soil is labeled as RFS in the caption but RFP in the figure.*

AC: Sorry. It is a typo. We have edited the figure **(fig.16)**.

**General and specific comments by anonymous referee 3 (Reviewer 3).**

*RC: The manuscript Discovery and characterization of submarine groundwater discharge in the Siberian Arctic seas: A case study in Buor-Khaya Gulf, Laptev Sea coauthored by Charkin et al. presents the first*

*evidences of the presence of SGD along the Eurasian Arctic margin. In my opinion the data provided in this manuscript is really interesting and will be the first step for other new studies. The paper is well written and the arguments are clear and well presented, although some parts of the text are difficult to follow. I recommend the publication of the manuscript after include some comments to the text. I have two major comments to the manuscript:*

*Why the study is so descriptive? The text describes the distribution of short-lived Ra isotopes and Rn but does not go deeply in the use of these radionuclides as a traces to estimate SGD fluxes or transit times.*

AC: Thank you for the overall support and for this specific comment.  We have now in the revised ms, incorporated a quantitative estimation of SGD discharge and transit times using short-lived Ra isotopes. Following the reviewers encouragement, we have now added an estimate of SGD discharge 1.7 x 10$^6$ m3 d-1 **(p.1, line 32; p.12, line 37; p.15, line 1)** and transit times 3.2 – 1.5 days **(p.12, line 23-31)** around the SGD place, while stating clearly all assumptions. Discharge of the subpermafrost groundwater from Kharaulakh hydrogeological massif through the talik area were calculated on excess 224Ra activities using a Ra mass balance model (Moore 1996; Burnett and Dulaiova 2003; Null et. al., 2012, 2014). In order to calculate the growth of the water mass "radium ages," we used the equation proposed by Moore (2000). As a result, there will be two new chapters **(chapter 2.5.2, p.6; chapter 2.2.3, p.7)** in the methods section and one in the discussion section **(chapter 3.5, p.11)**.

*RC: The second comment is for me the one that I can not understand. Where is the long-lived Ra data? I guess the authors have them and maybe they want to use in another manuscript. I feel that the long-lived Ra data will help to understand SGD sources and discharge processes. For example, recently Rodellas et al (GCA, 2017) have published how the combination of short- and long-lived Ra isotopes can be used to distinguish sources of submarine groundwater discharge: fresh groundwater vs seawater recirculation through sediments that is the case of part of the study.*

AC: We agree with this comment, but we still have not obtained the results of gamma spectrometry for the long-lived Ra isotopes for the wintertime (see our detailed response to a similar question by Rev. 1). However, in the final version of the manuscript we have included data on long-lived isotopes in the summertime **(p.9, line 26-42; p.10, line 1-4; fig.10; table 2)**:

Regarding methods for Ra isotopes, this will be included in the revised ms.  Briefly, in the shorebased/home laboratory, Ra was leached from the fibre with hot 6N HCl, coprecipitated as BaSO4 and counted with gamma spectroscopy for 226Ra and 228Ra (Moore, W.S., 1984. Radium isotope measurements using germanium detectors **(p.6, line 19)**. Nuclear Instruments and Methods in Physics Research 223, 407-411).

*RC: How you can discard that the short-lived Ra isotopes are not coming from recirculation or sediment resuspension?*

AC: This is a very important comment and we have now included  (in section 3.3) the long lived isotope data of the summertime cruise and an additional figure of radium isotopes versus salinity as supporting argument that the high activities of short lived Ra isotopes near station 1529 are due to SGD and not to resuspension **(p.9, line 26-42; p.10, line 1-4; fig.10; table 2)**. Besides, according to our multi-year winter data, the resuspension effect is negligible during wintertime (Charkin et al., BG, 2011)

**Some other technical corrections are:**

*RC: Do you have any sample for Ra-isotopes and Rn that can be compared between winter and summer?*
*o Line 35 pag 5. Do you need the two dots after Radiometer?*

AC: Unfortunately, we do not have such data.

*RC: Review the text because sometimes you write "kilometers" or "meters" and others you write "km" and "m".*

AC: Yes, we apologize for not being consistent in use of these jnits. We have checked the text carefully and streamline this

*RC: I suggest rewriting the sentence of the Line 20 Pag 6 as: These TEM results agree well with data provided by Imaev et al (2004) for this region.*

AC: Ok, we have done **(p.7, line 32)**.

*RC: Line 22 Page 6. Remove a space between 162, and 5 meters.*

AC: Ok, we have done **(p.7, line 33)**.

*RC: 3.2 Features of the thermohaline water structure and SGD fate. In this chapter authors explained the features of the thermohaline water structure but there is not any comparison or relation with SGD. Is this a correct tittle?*

AC: Yes, we agree: that is not correct. It have changed to Features of the thermohaline water structure **(p.8, line 16).**

*RC: In the chapter 3.3 the units of Ra-223, Ra-224 and Rn are wrong. It is "dpm ˚u100L-1" or "dpm/100L". Check the document and Figures.*

AC: We are sorry, this is a typo. It should of course be dpm/100L or dpm 100L-1. It have edited to dpm 100L-1.

*RC: Line 25 Page 8. Add a space between the numbers and "m" as you do for "km" or other units.*

AC: Ok, we have done **(p.9)**.

*RC: Line 10 to 13 Pag 8. Maybe the 228/226 AR provides information about the SGD*
*fluxes.*
AC: Yes, you are right. We included to the manuscript the summer data on long-lived isotopes which supporting argument that the high activities of short-lived Ra isotopes near station 1529 are due to SGD. **(p.9, line 26-42; p.10, line 1-4; fig.10; table 2)**

*RC: Line 10 page 9. Maybe this is a possible explanation, but what about the increase due to resuspension processes? Is it a possibility? Maybe the long-lived Ra isotopes can help you to understand the origin of this high 224/223 AR.*

AC: Yes, you are right, this could be an explanation for summer time, but in under ice conditions at a relatively large depth and with a low winter Lena River discharge, resuspension is very unlikely. This is also indicated by the very low concentration of suspended particulate matter in the water for this time (Charkin et al. BG, 2011). Moreover, we included to the manuscript the summer data on long-lived isotopes which supporting argument that the high activities of short-lived Ra isotopes near station 1529 are due to SGD and not to resuspension **(p.9, line 26-42; p.10, line 1-4; fig.10; table 2)**.

*RC: Line 33 page 9. Correct the 100L-1.*

AC: We have corrected **(p.10, line 32)**.

*RC: Line 20 page 10. Can you explain why this water is saltier?*

AC: This water is saltier because this water was formed as the result of multi-year freezing of rocks and the simultaneous concentration of salts (Pinneker, 1983; Romanovskii, 1983), resulting in elevated levels of total dissolved solids in the cryogenic groundwater **(p.13, line 21)**.

*RC: Line 38 page 10. Here you notice: The fact that the summertime and wintertime SGD springs were found on a line parallel to the fault once again points to the connection between the tectonogenic talik and the SGD. Here you talk about summertime and wintertime SGD springs. Why the seasonality is not shown in the text?*

AC: The seasonality is not shown in the text, because the summer data is very limited for SGD place.

*RC: Table 1. Add the uncertainties to the 224/223 AR and review the significant figures of the Ra-223 and Ra-224.*

AC: Ok, we have done **(table 1)**.

*RC: What is the meaning of the equation of the upper plot?*

AC: We have altered this piece of the text by removing both the linear mixing lines (because short-lived isotopes can be highly modified by decay in addition to mixing) and the equation.

*RC: I guess there is a mistake in the Figure referred to CSW in the plot and RFS in the caption.*

AC: Yes, you are right. It is a typo. We have corrected **(fig.16)**. Thank you.

**Discovery and characterization of submarine groundwater discharge in the Siberian Arctic seas: A case study in the Buor-Khaya Gulf, Laptev Sea**

Alexander N. Charkin[1,2], Michiel Rutgers van der Loeff[3], Natalia E. Shakhova[2,4], Örjan Gustafsson[5], Oleg V. Dudarev[1,2], Maxim S. Cherepnev[2], Anatoly N. Salyuk[1], Andrey V. Koshurnikov[6], Eduard A. Spivak[1], Alexey Y. Gunar[6], Alexey S. Ruban[2], Igor P. Semiletov[1,2,4]

[1]Pacific Oceanological Institute (POI), Far Eastern Branch of Russian Academy of Sciences Russian Academy of Sciences (FEBRAS) , Vladivostok, Russia
[2]National Research Tomsk Polytechnic University, Russia
[3]Alfred-Wegener Institute, Helmholtz Center for Polar and Marine Research, Bremerhaven, Germany
[4]International Arctic Research Center (IARC), University of Alaska, Fairbanks, USA
[5]Dept. of Environmental Science and Analytical Chemistry, and the Bolin Centre for Climate Research, Stockholm University, Stockholm, Sweden
[6]Moscow State University, Russia

*Correspondence to:* Alexander N. Charkin (charkin@poi.dvo.ru)

**Abstract.** It has been suggested that increasing terrestrial water discharge to the Arctic Ocean may partly occur as submarine groundwater discharge (SGD), yet there are no direct observations of this phenomenon in the Arctic shelf seas. This study tests the hypothesis that SGD does exist in the Siberian-Arctic shelf seas, but its dynamics may be largely controlled by complicated geocryological conditions such as permafrost. The field-observational approach in the southeast Laptev Sea used a combination of hydrological (temperature, salinity), geological (bottom sediment drilling, geoelectric surveys), and geochemical ($^{224}$Ra, $^{223}$Ra, $^{228}$Ra and $^{226}$Ra) techniques. Active SGD was documented in the vicinity of the Lena River delta with two different operational modes. In the first system, groundwater discharges through tectonogenic permafrost talik zones was registered in both winter and summer. The second SGD mechanism was cryogenic squeezing out of brine and water-soluble salts detected on the periphery of ice hummocks in the winter. The proposed mechanisms of groundwater transport and discharge in the arctic land-shelf system is elaborated. Through salinity versus $^{224}$Ra and $^{224}$Ra/$^{223}$Ra diagrams, the three main SGD-influenced water masses were identified and their end-member composition was constrained. Based on simple mass balance box models, discharge rates at site in the submarine permafrost talik zone were $1.7 \times 10^6$ m$^3$d$^{-1}$ or 19.9 m$^3$s$^{-1}$, which is much higher than the April discharge of the Yana River. Further studies should apply these techniques on a broader scale with the objective of elucidating the relative importance of the SGD transport vector relative to surface freshwater discharge for both water balance and aquatic components such as dissolved organic carbon, carbon dioxide, methane, and nutrients.

**1 Introduction**

The Arctic system constitutes a unique and important environment with a central role in the dynamics and evolution of the earth system. Global warming has regional effects on the Arctic, including on all cryospheric features. Energy and water fluxes shape the regional temperature regime, which is a primary factor in determining the physical state

Добавлено примечание ([11]): We have reworked this sentence according to the comment of the first reviewer

Добавлено примечание ([12]): We have added an estimate of SGD

[revised manuscript text omitted]

**Добавлено примечание ([15]):** We have added this information to the methods according to the comment of the second reviewer

**Добавлено примечание ([16]):** We have added this information to the methods according to the comment of the second reviewer

**Добавлено примечание ([17]):** New chapters about the calculations of SGD residence time

[revised manuscript text omitted]

**Добавлено примечание ([19]):** We have included data on long-lived isotopes in the summer

**Добавлено примечание ([110]):** We have re-organized this section according to the comment of the second reviewer. Now the first endmember descriptions, and then the discussion in the context of the two figures

The close $^{224}$Ra/$^{223}$Ra activity ratio of these two water masses (Table 1, Fig. 12b) is explained by the large participation of SW in TGW formation after GW discharge. As the result of this mixing, the highest activities of $^{224}$Ra and $^{223}$Ra were measured in this water; these isotopes are excellent tracers of SGD (Moore and Arnold, 1996; Rama and Moore, 1996; Charette et al., 2008). The $^{224}$Ra/$^{223}$Ra ratio in the discharge location was 29, which also points to a mixing of GW and bottom SW (GW $^{224}$Ra/$^{223}$Ra=26; SW average of $^{224}$Ra/$^{223}$Ra=35). However, this is not a feature of SGD at the locations of the station 1504 and 1505 bottom horizons. First of all, the samples were taken 1m from the bottom (see methods) and it was precisely at these stations that the low-temperature fluid was closest to the bottom (Fig. 7); second, the $^{224}$Ra/$^{223}$Ra ratio at these stations was less than at station 1529, with similar values of salinity, which shows $^{224}$Ra decay with time after SGD. The other situation is characteristic of the bottom horizon at station 1520; here SGD was also not discovered by temperature. However, $^{224}$Ra and $^{223}$Ra activity and the $^{224}$Ra/$^{223}$Ra ratio were comparable with those measured at station 1529 (Table 1, Fig. 8). Additionally, a shifting of isohalines was observed at station 1520 similar to that observed at station 1529 (Fig. 7), which points to mixing. This circumstance can be explained in two ways; either the source of the SGD is located close to station 1520, or the discharge has a pulsating character.

The $^{224}$Ra/$^{223}$Ra activity ratio in the SW (bottom water) is higher than in the TGW, BW, and MGRSW. This $^{224}$Ra/$^{223}$Ra activity ratio is possibly explained by formation of a high density near – bottom winter nepheloid layer caused by flocculation of humic substances and mineral particulates due to a lack of active mixing. Sinking Th isotopes scavenged from surface horizons of the water column accumulate on the organic and mineral particles in the bottom nepheloid horizon. Because of the much greater activity of $^{228}$Th (half-life = 1.9 years) with respect to $^{227}$Th (half-life = 18.9 days) the $^{227}$Th decays faster than the $^{228}$Th; as a result, after 5 months of winter the $^{228}$Th/$^{227}$Th and $^{224}$Ra/$^{223}$Ra ratios increase.

The river endmember is well identified in the theoretical mixing diagram; some displacement of station 1531 seen in Fig. 12a is explained by the decay of $^{224}$Ra. This is indicated by a decrease in the $^{224}$Ra/$^{223}$Ra activity ratio at this station.

We detected MGRSW in the surface plume at station 1520 ($^{224}$Ra activity is 1.3 times higher here than at the surface of the neighboring stations 1529 and 1504), and in the waters on the periphery of the ice hummocks (called *stamukha* in Russian) in the shallows around Cape Muostakh. This MGRSW features higher $^{224}$Ra activity compared to BW, as well as higher salinity and a higher $^{224}$Ra/$^{223}$Ra activity ratio. In addition, station 1501 lies above the mixing line of the RW and SW endmembers, which also indicates the involvement of GW in the formation of this water mass.

Considering all the above data together, we have constructed a scheme for mixing water masses at the SGD site (Fig. 13). At the first stage, discharge GW is mixed with inflow bottom SW. After mixing, the TGW occupies an intermediate position between the bottom SW and surface RW-BW water according to its density. As a result, and because of the absence of wind mixing beneath the ice, a highly stratified water mass is formed (Fig.5). Due to the existence of a natural barrier in the form of the delta edge and the Lena River runoff, the distribution of TGW to the west is limited; it is found primarily east and south of the investigated area (Figs. 6, 7). The constant GW flow contributed to the formation of a small field of higher salinity water in comparison with the surrounding waters on the surface, slightly shifted to the east under the influence of Lena River runoff.  We also identified these waters as MGRSW (Fig.12, 13).

**3.5 Estimate of SGD in the open talik zone**

Estimate of SGD in the open talik zone calculated using Eq. (2) and from ex$^{224}$Ra activities and the box (prism) model is represented in Fig. 14. *Vbox* is the volume of the box which is shaped like a prism, defined by corners with known

Добавлено примечание ([111]): We have corrected this section according to the comment of the second reviewer.

Добавлено примечание ([112]): New chapters about the SGD discharge and residence time

[revised manuscript text omitted]

**Добавлено примечание ([113]):** We have removed the radon data since we do not use them, added density data for box model and uncertainties to the 224/223 AR.

[revised manuscript text omitted]

**Добавлено примечание ([120]):** We have corrected this figures according to the reviewers comment

---

## Author Response (AR2)

Author response to Handling Editor

We thank Nina Kirchner for very constructive comments, which have been useful in our revisions of the manuscript. Below, Editor Comment is listed first, followed by our response and a description of resulting edit. Author comments are marked below as AC.

*EC: Caption Figure 2: Add source for the background map in panel a. I suspect it is IBCAO?*
*Panel b: Put station numbers in parantheses. ".. between winter (no. 1529) and summer (no. 103)*
*stations". Panel c: What is TS? All abbreviations must be explained.*

AC: Thank you. We have edited accordingly

*EC: Caption Figure 4: You should refer to Fig. 2 for an overview map. Also, it would be good if the tower symbols used in the figure would be explained in the caption. Panel b: Add the names to the drilling cores: "of the drilling cores 1D-15, ...".*

AC: Thank you. We have edited accordingly

*EC: Caption Figure 6: Why is there a white area in all the six panels which show the enlarged region? I think this needs to be explained, and mentioned in the caption.*

AC: Thank you. We have painted a white area in all the six panels, which show the enlarged region. Initially, it seemed to us that it was better not to paint this region, because there were not enough stations. However, in principle, since this region inside the already painted all research area, we finally decided to paint it.

*EC: Figure 7: The map in the uppermost panel is still too small. Further, why is there a white gap in all the panels? This should be explained in the caption.*

AC: Yes, you are right the map is still too small, so we decided to delete it. Below we have added how you requested that the location of transects T1 and T2 is shown in Figure 2, so we do not need this map anymore. We have explained a white gap in the caption.

*EC: Caption Figure 7: Add that the location of transects T1 and T2 is shown in Figure 2. Also, the text mentions that "mushroom-like structures" can be seen. I can not see such structures, so an explanation in the caption and an arrow (or any other indicator) should be included in the panels whereever these structures appear.*

AC: We have added information about the location of transects T1 and T2. In addition, we have added yellow cell in the figure for a "mushroom-like structure" identification and explain it in the caption.

*EC: Figure 15: The red text in panel a can hardly be read. Please make it more readable.*

AC: Thank you. We have edited accordingly

*EC: Further, please type equations 1 and 2. Right now, it looks that they are pasted in as images.*

AC: Yes, you are right it was images. Now we have changed to the text format.

*EC: Finally, please check the language once more. Specifically, at two occasions, "region" should be used instead of "territory" (line 19, first page of chapter 3), and instead of "district" (line 41, sec 3.2).*

AC: Thank you. We have edited accordingly

**Thank you for your valuable comments, which help to improve our manuscript.**

[revised manuscript text omitted]
 2 and we added this information to the caption, so we do not need this map anymore. And we have added yellow cell for a "mushroom-like structure" identification.

**Добавлено примечание ([19]):** We have added information about the location of transects T1 and T2. In addition, we have added an explanation for a "mushroom-like structure" identification and white gap existence.

[Figure]

**Figure 8: Winter distributions of ex$^{224}$Ra in surface (a) and bottom (b) water. At the shallow stations one value is shown for both horizons.**

[Figure]

**Figure 9: Vertical profiles of the winter water column.**

[Figure]

**Figure 10: Activities of [224]Ra, [226]Ra, and [228]Ra as function of salinity during the summertime cruise. Bottom water activities at station 103 (12 m, black) and Y (10 m, blue) are the only ones that stand out for all three isotopes. Figure made with Ocean Data View (Schlitzer, R., Ocean Data View, http://odv.awi.de, 2017)**

[Figure]

**Figure 11: Summer (September 2013) vertical distribution of ex$^{224}$Ra in the Lena Delta. Sta Y (72°N, 130°E) is included in transect T1.**

[Figure]

**Figure 12: Salinity vs. $^{224}$Ra (a) and $^{224}$Ra/$^{223}$Ra (b) in the winter.**

a) The three main water-mass endmembers are identified in this diagram: River water (RW - light green dots), seawater (SW - blue dots), and transformed ground water (TGW - red dots), and two derivatives from the main water mass because of mixing: mixed river and seawater (BW - dark green dots), and mixed RW, GW, and SW (MRGSW - yellow dots). Black arrows are theoretical decay lines.

b) Similar to the mixing diagram panel (a) but using the $^{224}$Ra/$^{223}$Ra ratio. The straight line represents the theoretical mixing line derived from the major radium sources. Mid: middle of the water column, bott: bottom.

[Figure]

**Figure 13: The scheme of water masses mixing near the SGD location. The large black arrows show direction of endmembers movement, small light blue arrows show direction of GW flow movement, black spirals show mixing.**

[Figure]

**Figure 14: Scheme of box model mass-balance mixing for SGD in the studied area. Red ellipse - the proposed SGD location. Long red arrows show the direction of TGW plume movement, numbers in red circles show the residence time (days), blue triangle indicates the base of the prism for the box model.**

**Добавлено примечание ([110]):** We have edited this figure accordingly Editor Comment.

[Figure]

**Figure 15: The mechanistic scheme of recharge, transport, and discharge of pressurized submarine GW.**

**a) The general vertical scheme of recharge, transport, and discharge of pressurized submarine GW.**

5 **b) The spatial scheme of GW transport in the Kharaulakh hydrogeological massif.**

**c) The vertical scheme of GW transport and submarine discharge instead of tectonogenic talik based on TEM and drilling data. 1) Metamorphic bedrock of Pre-Upper Cretaceous complexes, 2) substratum, 3) sands, 4) aeolian Quaternary material, 5) recent bottom sediments, 6) direction of GW movement, 7) Geothermal flux, 8)**

10 **SGD, 9) place and number of TEM measurements.**

[Figure]

**Figure 16: Scheme of CSB and expulsion of water-soluble salts from the ice-formation zone into warmer, unfrozen parts of the sediments. RFS – recently frozen soil. RP- relict permafrost.**